# CORRECT: Condensed Error Recognition via Knowledge Transfer in Multi-agent Systems

Yifan Yu [1]  Moyan Li [2]  Shaoyuan Xu [2]  Jinmiao Fu [2]  Xinhai Hou [3]  Fan Lai [1]  Bryan Wang [2]

## Abstract

Multi-agent systems (MAS) are increasingly capable of tackling complex real-world tasks, yet their reliance on inter-agent coordination, tool use, and long-horizon reasoning makes error recognition particularly challenging. Minor errors can propagate across agents, escalating into task failures while producing long, intertwined execution trajectories that impose significant costs for both human developers and automated systems to debug and analyze. Our key insight is that, despite surface differences in failure trajectories (e.g., logs), MAS errors often recur with similar structural patterns. This paper presents *CORRECT*, the first lightweight, training-free framework that leverages an online cache of distilled error schemata to recognize and transfer knowledge of failure structures across new requests. This cache-based reuse allows LLMs to perform targeted error localization at inference time, avoiding the need for expensive retraining while adapting to dynamic MAS deployments in subseconds. To support rigorous study in this domain, we also introduce *CORRECT*-Error, a large-scale dataset of over 2,000 annotated trajectories collected through a novel error-injection pipeline guided by real-world distributions, and further validated through human evaluation. Experiments across seven diverse MAS applications show that CORRECT improves error localization up to 19.8% over existing advances while at near-zero overhead. Our code is publicly available at: https://github.com/UIUC-MLSys/CORRECT.

Part of this work was done while Yifan Yu was an intern at Amazon. [1]University of Illinois Urbana-Champaign [2]Amazon [3]University of Michigan. Correspondence to: Fan Lai <fanlai@illinois.edu>, Bryan Wang < brywan@amazon.com>.

*Proceedings of the 43rd International Conference on Machine Learning*, Seoul, South Korea. PMLR 306, 2026. Copyright 2026 by the author(s).

## 1. Introduction

Multi-agent systems (MAS) have demonstrated success in various domains, including software development (Qian et al., 2023; Hong et al., 2024; Zhang et al., 2024), scientific research (Lu et al., 2024), web navigation (Zhou et al., 2023), and general-purpose task automation (Wu et al., 2024; Wang et al., 2025). By orchestrating multiple specialized agents, MAS can tackle challenges beyond the reach of single-agent systems (SAS) that rely on single LLMs for task solving.

However, as MAS increasingly scale in both complexity (e.g., sophisticated interactions with other agents and tools) and deployments, decisive error recognition that pinpoints the precise agent and step that first triggered the failure (Zhang et al., 2025) has been a fundamental challenge (Pan et al., 2025). Unlike SAS where failures can be traced to a single faulty output, MAS failures often emerge from cascading effects across agents: an error-prone step by one agent can propagate through downstream interactions, ultimately causing task failure (Gao et al., 2025). Efficient decisive error recognition is essential for reliable MAS deployments, sustaining service quality, and guiding operational management such as safe agent upgrades, targeted restarts, and service monitoring (Epperson et al., 2025).

Unfortunately, decisive error recognition in MAS remains open-ended due to three fundamental obstacles: (1) *Generality*: MAS span a wide spectrum of applications, and even within an application, requests can exhibit drastically different error patterns, making it difficult to design methods that generalize. Existing advances (Zhang et al., 2025) often resort to LLM-as-a-judge methods for generality, yet achieve ≤10% accuracy. Recent efforts (Ge et al., 2025) to improve accuracy through fine-tuning not only hurt generalization, but fall short in (2) *Data Efficiency*: obtaining labeled data in error recognition is notoriously expensive and inherently ambiguous. For example, in the WHO&WHEN dataset (Zhang et al., 2025), annotators spent over 30 expert hours labeling fewer than 200 trajectories, yet disagreement rates exceeded 50%. This makes any training-based approaches ineffective without voluminous data; and (3) *Computation Efficiency*: even with sufficient data, tuning LLMs for error recognition may be impractical to catch up with deployments where new error types arise continuously and sporadically, e.g., new

attacks in cloud AIOps (Wang et al., 2025).

In this paper, we first notice that failures in MAS tend to re-cur with similar structures across requests (§2.2). Because a MAS application often relies on the same role specifications, orchestration rules, tool APIs, or verification policies, di-verse requests often funnel into common decision skeletons. Our real-world analysis of WHO&WHEN dataset proposed in Zhang et al. (2025) shows that over 80% of failure tra-jectories have at least one counterpart with $\geq 0.8$ semantic similarity in error logs. This suggests that error knowledge can be systematically *distilled, cached, and reused*. How-ever, naive approaches, such as in-context learning (ICL) that insert prior trajectories as in-context exemplars, quickly break down: logs can exceed 32,000 tokens (Yang et al., 2025; Dubey et al., 2024), contain low-entropy noise, and even underperform zero-shot baselines (§2.2).

In this paper, we present *CORRECT* (COndensed eRror RECognition via knowledge Transfer), a novel framework that elevates *decisive error recognition* to a first-class sys-tems problem for practical and reliable MAS. *CORRECT* automatically distills prior failures into compact, reusable error schemata that encode their core signatures, triggering contexts, and propagation patterns. At runtime, when a fail-ure occurs, *CORRECT* retrieves and instantiates the most relevant schemata to diagnose the new trajectory, enabling accurate, training-free error recognition in dynamic online environments. We summarize our contributions as follows:

- **CORRECT: the first schema-guided detector.** We introduce *CORRECT*, the first framework that systemat-ically distills recurrent MAS failures into compact error schemata and reuses them for decisive error recognition. In contrast to LLM-as-a-judge approaches that sacri-fice accuracy for generality, and fine-tuning approaches that incur high data and compute costs, *CORRECT* transfers error knowledge across requests at inference time via schema retrieval. This design improves step-level localization accuracy by up to 20 points over prior work (Zhang et al., 2025), while remaining training-free and lightweight for large-scale online deployment.

- *CORRECT*-**Error: a large-scale, and high-fidelity dataset for MAS error recognition.** To address the lack of reliable evaluation data, we construct *COR-RECT*-Error, a benchmark of over 2,000 multi-agent tra-jectories with fine-grained, step-level error annotations. *CORRECT*-Error is generated via a novel error-injection pipeline guided by real-world failure distributions, pro-ducing data that combines scalable coverage with the realism of natural MAS failures. Extensive human val-idation shows strong alignment between synthetic and expert-labeled errors. Beyond enabling rigorous eval-uation of *CORRECT*, *CORRECT*-Error establishes a reusable and extensible benchmark for the community.

These contributions advance the state of the art in MAS error recognition and establish a foundation for building more reliable, interpretable, and scalable multi-agent systems. We will release the datasets and benchmarks.

## 2. Background and Motivation

### 2.1. Decisive Error in Multi-Agent Systems

Task failures in MAS often arise from specific *decisive errors* that, once committed, make successful task com-pletion impossible. We formalize this notion following prior work (Zhang et al., 2025). Consider a MAS exe-cuting a trajectory $\tau = \{(a_1, s_1), (a_2, s_2), \ldots, (a_T, s_T)\}$, where agent $a_i$ performs step $s_i$. The outcome of the trajectory is denoted by $\mathcal{R}(\tau) \in \{0, 1\}$, with 1 for suc-cess and 0 for failure. A step $(a_k, s_k)$ in a failed trajec-tory $\tau$ is a *decisive error* if replacing it with a correct al-ternative $\tilde{s}_k$ would change the outcome to success. The earliest decisive error is $(a^*, s^*) = \min_{k \in \mathcal{D}(\tau)} k$, where $\mathcal{D}(\tau) = \{ k : \mathcal{R}(\tau) = 0 \wedge \mathcal{R}(\tau_{[s_k \to \tilde{s}_k]}) = 1 \}$, and $\tau_{[s_k \to \tilde{s}_k]}$ denotes the modified trajectory where step $s_k$ is replaced.

Intuitively, a decisive error is **the earliest step** whose cor-rection flips a trajectory's outcome from failure to suc-cess. Identifying such errors is fundamental: Unlike coarse-grained error recognition, which merely flags failed trajec-tories, decisive error recognition localizes the precise *agent* and *step* that initiated failure. This fine-grained attribution enables targeted interventions, such as tuning role specifica-tions, refining orchestration logic, or upgrading individual agents, without costly overhauls of the entire system.

### 2.2. Motivations of *CORRECT*

Automated decisive error recognition in MAS has primarily followed two directions: LLM as a judge and fine-tuning specialized LLMs. Both exhibit fundamental limitations in accuracy, generality, and efficiency.

**Limitations of Existing Advances.** LLM-as-a-judge methods were initially designed to rate the quality of LLM outputs, achieving up to 80% agreement with human pref-erences (Zheng et al., 2023). Zhang et al. (2025) extended this paradigm to MAS error attribution, introducing three variants: (i) *all-at-once*, which provides the full error log to the LLM and asks it to identify the error step; (ii) *step-by-step*, which incrementally reveals the trajectory and checks errors at each step; and (iii) *binary search*, which recursively partitions the log to localize the error. However, as MAS be-comes more complex, which elongates the failure trajectory, these methods lose diagnostic precision (Figure 7): on the WHO&WHEN dataset (Zhang et al., 2025), Qwen-2.5-7B achieves only 3.5% step-level accuracy.

Fine-tuning-based approaches, whether supervised or rein-

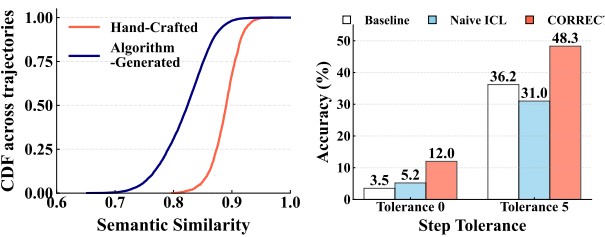

*(a)* Failure trajectories exhibit high semantic similarities (WHO&WHEN dataset).

*(b)* Performance of naive ICL and our method (WHO&WHEN dataset).

*Figure 1.* Analysis of failure traces on the WHO&WHEN dataset.

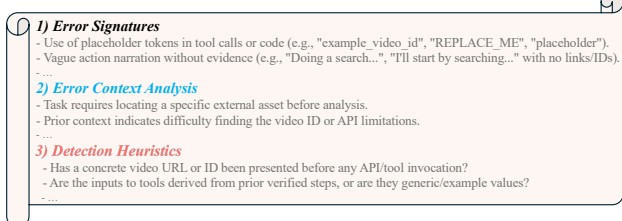

*Figure 2.* Example of an error schema generated on the WHO&WHEN dataset.

forcement learning–based, face substantial efficiency and expense challenges. Their success hinges on large-scale, high-quality labeled datasets. Yet annotating MAS failures is prohibitively expensive: annotators must disentangle long, interdependent interactions across agents and tools, taking 30 expert hours for annotating fewer than 200 trajectories. Worse, error trajectories vary across applications and requests, making it infeasible to generalize across settings.

**Pervasive Error Similarity Yet Hard to Reuse.** Despite these challenges, our analysis of the WHO&WHEN dataset (Figure 1a) reveals that more than 80% of failed requests share a semantic (cosine) similarity above 0.8, measured via BERT-based embeddings. This reveals an underexploited opportunity: error knowledge recurs and could, in principle, be reused across requests. A natural attempt is to adopt in-context learning (ICL), retrieving and appending similar trajectories to guide error recognition. However, our experiments (Figure 1b) show that such a strawman approach even degrades accuracy, due to (i) *extreme trajectory length*: 17% of trajectories exceed 32K tokens (details in Appendix A.2), surpassing the context length of most LLMs (e.g., Qwen3 (Yang et al., 2025)); and (ii) *low signal-to-noise ratio*: execution trajectories interleave request-specific details and tool outputs with the true error-inducing steps, diluting critical information.

## 3. CORRECT: Condensed Error Recognition via Knowledge Transfer

Our observations call for a novel approach that can *systematically reuse structural error knowledge* without overwhelming context or succumbing to noise, while remaining general, data-efficient, and lightweight for real-time MAS deployment. To these ends, we introduce *CORRECT* (**CO**ndensed e**R**ror **REC**ognition via knowledge **T**ransfer), the first framework that distills past errors into compact error schemas and adaptively applies them for accurate decisive error recognition, entirely without training. We next introduce how CORRECT achieves this via three phases: (1) *Offline schema extraction*, (2) *Online schema-guided*

*error recognition*, and (3) *Dynamic schema management*.

### 3.1. Error Schema Extraction

Given an annotated error trajectory $\mathcal{T} = \{(a_i, s_i, r_i)\}_{i=1}^n$, where $a_i$ denotes the agent at step $i$, $s_i$ represents the step content, and $r_i$ is the corresponding result, along with the identified error at step $s_e$ and error reason $r_e$, *CORRECT* generates an error schema $\mathcal{S}$ capturing (Figure 2): (1) *Error Signatures* $\Sigma$: characteristic patterns such as agent actions, interaction sequences, and key behavioral markers; (2) *Error Context Analysis* $\mathcal{C}$: detailed analysis of the conditions that led to the error, including agent states, task progress, and environmental factors; and (3) *Detection Heuristics* $\mathcal{H}$: actionable rules and guidelines for identifying similar errors.

To minimize human efforts, *CORRECT* leverages LLMs (e.g., GPT-5 or Qwen3) to generate error schemas. We discuss how to ensure the quality of the schema in Section 3.2.

**Clustered Schema Extraction.** Even with LLMs, generating a schema for every trajectory can be costly, considering voluminous requests in practical MAS deployments. In fact, doing so is unnecessary because schema reuse often follows a long-tailed distribution: a small number of schemas are frequently reused, while most are rarely applied (§5). To exploit this, *CORRECT* performs the following offline procedure: (1) *Trajectory Clustering:* Failure trajectories are embedded semantically and clustered to group similar error patterns, and then (2) *Cluster-level Schema Generation:* One representative schema is generated per cluster, capturing the common error structure.

The number of clusters is selected using a data-driven criterion that balances schema compactness and diagnostic coverage. Let $\{\mathbf{e}_i\}_{i=1}^N$ denote the semantic embeddings of $N$ failure trajectories, and let $\mathcal{C}(K)$ be the clustering result with $K$ clusters. For each trajectory $i$, we compute its silhouette coefficient:

$$s(i) = \frac{b(i) - a(i)}{\max\{a(i), b(i)\}},$$

where $a(i)$ is the average distance between $\mathbf{e}_i$ and other points in the same cluster, and $b(i)$ is the minimum average distance to points in any other cluster. We define the average

silhouette score as $S(K) = \frac{1}{N} \sum_{i=1}^{N} s(i)$. We increase $K$ until the marginal gain in cohesion saturates, i.e., $S(K+1) - S(K) < \epsilon$, for a small threshold $\epsilon$. As shown in our cache-size ablations (§5.2), a relatively small set of schemas (often a few hundred) already reaches an accuracy plateau, confirming the effectiveness of our design.

### 3.2. Schema-Guided Error Recognition Online

Once a failure request requires diagnosis, we start with its trajectory $\mathcal{T}_{\text{target}}$ and retrieve the top-k relevant schemas from the cache via semantic similarity search (e.g., cosine similarity of embeddings):

$$\text{sim}(\mathcal{T}_{\text{target}}, \mathcal{S}_i) = \cos(\text{embed}(\mathcal{T}_{\text{target}}), \text{embed}(\mathcal{S}_i))$$

With retrieved schemas, we prompt the LLM to perform error recognition by instantiating the schemas in the context of the target trajectory. These schemas, together with the trajectory and lightweight adaptation instructions, are passed to an LLM for diagnosis. Formally, the LLM receives $(\mathcal{T}_{\text{target}}, \{\mathcal{S}_j\}_{j=1}^{k}, \text{prompt}_{\text{detect}})$ as input prompt and produces the error recognition result:

$$\text{result} = \text{LLM}_{\text{detect}}(\mathcal{T}_{\text{target}}, \{\mathcal{S}_j\}_{j=1}^{k}, \text{prompt}_{\text{detect}}).$$

By leveraging schemas as condensed expert knowledge, this schema-guided inference directs the LLM's attention toward salient failure patterns, avoiding the cost and noise of processing full historical trajectories.

**Adaptation with Schema Expansion and Distillation.** *CORRECT* maintains an effective schema cache through two complementary mechanisms. First, *schema expansion*: when user feedback confirms successful recognition, *CORRECT* leverages the ground truth label from the user to generate and cache a new error schema following 3.1. Priority is given to trajectories with low similarity to existing schemas (i.e., $\text{sim}(\mathcal{T}_{\text{new}}, \mathcal{S}_i) < \delta$ for all cached schemas), ensuring the cache captures diverse error patterns rather than redundant ones. Second, *schema distillation*: expansion alone may yield suboptimal quality, and frequently accessed error schemas (cache hits $> \theta_{\text{hot}}$) may benefit from further refinement. In such cases, *CORRECT* generates multiple candidate schemas and replays them against prior trajectories to select the discriminative one with the highest accuracy. Together, expansion ensures coverage for novel errors online, while distillation preserves cache efficiency by retaining only high-quality, discriminative schemas.

In practical deployments, each schema is typically reused across voluminous requests, so the overhead of both operations is largely amortized: Our evaluations show that tens of schemas already achieve strong accuracy for thousands of request errors (thus amortized overhead in generating one

---

**Algorithm 1** CORRECT Framework

**Require:** Annotated trajectories $\{(\mathcal{T}, s_e, r_e)\}$; target $\mathcal{T}_{\text{target}}$
**Ensure:** Error recognition result $(a^*, s^*, c)$

1: **Offline Schema Extraction** (Sec 3.1)
2: Cluster trajectories by semantic similarity
3: **for each** annotated trajectory $(\mathcal{T}, s_e, r_e)$ **do**
4:    $\mathcal{S} \leftarrow \text{LLM}_{\text{extract}}(\mathcal{T}, s_e, r_e)$
5:    Apply filtering/distillation
6:    $\mathcal{C}.\text{put}(\mathcal{S})$
7: **end for**

8: **Online Schema-Guided Error Recognition** (Sec 3.2)
9: $\mathbf{e} \leftarrow \text{embed}(\mathcal{T}_{\text{target}})$
10: $\{\mathcal{S}_j\}_{j=1}^{k} \leftarrow \mathcal{C}.\text{search\_top\_k}(\mathbf{e})$
11: $\forall j : \mathcal{C}.\text{update\_access}(\mathcal{S}_j)$
12: $(a^*, s^*, c) \leftarrow \text{LLM}_{\text{detect}}(\mathcal{T}_{\text{target}}, \{\mathcal{S}_j\})$

13: **Dynamic Schema Management** (Sec 3.2)
14: **if** user feedback confirms successful recognition **then**
15:    **if** $\text{sim}(\mathcal{T}_{\text{target}}, \mathcal{S}_i) < \delta$ for all $\mathcal{S}_i \in \mathcal{C}$ **then**
16:      Extract ground truth label from data
17:      Distill new schema $\mathcal{S}_{\text{new}}$
18:      $\mathcal{C}.\text{put}(\mathcal{S}_{\text{new}})$
19:    **end if**
20: **end if**
21: **if** $\mathcal{C}.\text{access\_count}(\mathcal{S}_j) > \theta_{\text{hot}}$ **then**
22:    $\{\mathcal{S}'_i\}_{i=1}^{m} \leftarrow \text{LLM}_{\text{extract}}^{m}(\mathcal{T}_j, s_{e,j}, r_{e,j})$
23:    Evaluate by replaying on prior trajectories
24:    $\mathcal{S}^* \leftarrow \arg\max_{\mathcal{S}'_i} \text{accuracy}(\mathcal{S}'_i)$
25:    $\mathcal{C}.\text{replace}(\mathcal{S}_j, \mathcal{S}^*)$
26: **end if**
27: **return** $(a^*, s^*, c)$

---

schema is below 1%). Moreover, both designs achieve consistent improvements across settings (e.g., $> \theta_{\text{hot}}$), and with these two mechanisms in place, the schema cache adapts robustly over time under evolving workloads (§5.1).

Algorithm 1 summarizes *CORRECT* runtime detection. The offline stage builds an initial cache by distilling clean, representative schemas from annotated trajectories (Line 1-Line 7). At test time, the system retrieves the top-$k$ relevant schemas to guide the LLM detector toward the most likely error (Line 8-Line 12). The cache tracks how often each schema is used so it can update itself: new schemas are added when no good match exists, and frequently accessed schemas are refined when needed. This keeps the cache compact, accurate, and aligned with live error patterns.

## 4. *CORRECT*-Error: A Large-Scale Error Detection Benchmark

Existing efforts for trajectory-level error analysis are limited in both scale and diversity, and human annotation is costly and difficult to scale (§2.2). To bridge this gap and evaluate *CORRECT*'s effectiveness (§3), we introduce *CORRECT*-Error, a large-scale benchmark that faithfully reflects the distribution of natural errors in real-world MAS.

## 4.1. Bootstrap Error Synthesis Pipeline

In building *CORRECT*-Error, we develop a novel bootstrap methodology that uses a small set of human-annotated error trajectories as seeds for scalable error generation, blending realism with controllability. It follows a three-stage pipeline that distills human expertise into scalable synthetic data while preserving the structural and semantic integrity of real-world error patterns.

**Stage 1: Diverse Trajectory Collection.** We first generate a large corpus of successful multi-agent trajectories spanning heterogeneous tasks and domains. In parallel, we curate a smaller but high-quality set of human-annotated error trajectories. These human-labeled examples serve as reference exemplars of realistic failure dynamics, capturing both localized mistakes and their downstream propagation.

**Stage 2: Semantic Error Schema Matching.** Each successful trajectory is paired with its closest human-labeled error trajectory using semantic similarity measures that account for both high-level task goals and fine-grained agent interactions. This alignment ensures that the selected error schema is contextually aligned with the target trajectory, avoiding unrealistic mismatches. Then we use LLMs (e.g., GPT-5) to devise an error injection strategy that specifies (i) where in the target trajectory to introduce the error, and (ii) how to adapt the error pattern while preserving its semantics.

**Stage 3: Contextual Error Injection.** Following the injection strategy, we prompt GPT-5 that generated the original successful trajectory to introduce an erroneous action at the designated point. This guarantees consistency in linguistic and behavioral style while embedding a realistic failure. We provide the prompt used to generate natural, schema-consistent error injections in Appendix A.9.

## 4.2. Human-Alignment Analysis

Following our bootstrap pipeline (§4.1), we synthesized over 2,000 trajectories across seven datasets (Figure 3), yielding **12.3×** more data than WHO&WHEN with cost over 3 billion tokens using GPT-5 series models and GPT-4o series models based on Magnetic-One (Fourney et al., 2024) and AutoGen (Wu et al., 2024). The resulting benchmark spans diverse tasks, including multi-hop QA, common planning, mathematical reasoning, and scientific problem-solving. By leveraging limited human annotations as seeds, our novel pipeline generates diverse error scenarios at scale.

To rigorously evaluate the authenticity of our synthesized data, we conducted a human evaluation study with four independent expert labelers, totaling over 120 hours of annotation effort. Each labeler was presented with an equal mix of synthetically injected and human-annotated error

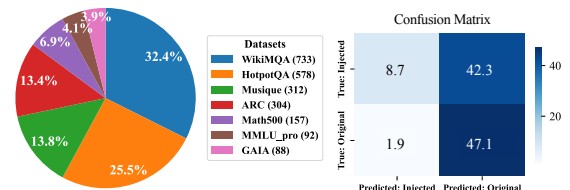

*Figure 3.* *CORRECT*-Error includes diverse tasks. The synthesized data preserves high realism, where human labelers frequently misclassified synthetic errors as genuine ones.

trajectories, without being informed of their origin. As shown in Figure 3, labelers struggled to distinguish between the two sources: 47.1% of synthetic trajectories were misclassified as human-labeled, while only 42.3% of genuine trajectories were correctly identified. This near-random classification performance, close to 50% in both cases, indicates that synthetic errors produced by our pipeline are effectively indistinguishable from real-world failures.

Beyond individual judgments, we observe strong inter-annotator agreement on perceived authenticity. In particular, 94.4% of synthetic trajectories were judged as genuine by at least two out of five labelers, and 52.9% received unanimous consensus. These results suggest that our injected errors consistently exhibit the structural and semantic characteristics of natural MAS failures, which also reinforces the effectiveness of error schema design in *CORRECT*. We provide a detailed human-alignment analysis, including confusion matrices and agreement statistics, in Appendix A.4.

Taken together, these findings validate that our error injection pipeline faithfully captures the nuanced failure patterns observed in real multi-agent systems. This methodology enables data generation that is *cheap* (fully automated and low-cost), *abundant* (scalable to millions of trajectories), *precise* (with unambiguous ground-truth error labels), and *realistic* (closely matching human-annotated error distributions), thereby providing a strong foundation for developing and evaluating effective MAS error recognition methods.

## 5. Experiments

We demonstrate that *CORRECT* achieves significant accuracy improvements (up to 20%) on WHO&WHEN and an average gain of 28.7% across MAS tasks (§5.1) on CORRECT-ERROR, all at zero training costs. *CORRECT* remains robust under distribution shifts arising from model updates, dataset variations, schema cache size, and the number of stored schemata (§5.2). We provide case-studies with illustrative examples and detailed analysis in Appendix A.11.

**Models and Tasks.** We evaluate *CORRECT* on both the human-annotated WHO&WHEN benchmark and our high-quality benchmark, CORRECT-ERROR, which spans

| Method | Model | Human-Crafted | | Algorithm-Generated | |
|---|---|---|---|---|---|
| | | Acc@0 | Acc@1 | Acc@0 | Acc@1 |
| LLM-as-a-Judge | Qwen-2.5-7b | 3.5 | 8.6 | 19.1 | 42.9 |
| | Qwen-3-30b | 1.7 | 5.2 | 15.1 | 42.9 |
| | Qwen-3-80b | 6.9 | 8.6 | 21.4 | 47.6 |
| | Llama-8b | 1.7 | 3.5 | 3.2 | 15.9 |
| | DeepSeek-R1 | 3.5 | 17.2 | 23.8 | 54.0 |
| | Gemini-2.5-flash | 5.2 | 13.8 | 31.8 | 56.0 |
| | Gemini-2.5-pro | 5.2 | 12.1 | 25.4 | 50.0 |
| | GPT-4o-mini | 3.5 | 12.5 | 12.7 | 39.7 |
| | GPT-4o | 3.5 | 10.3 | 18.3 | 50.0 |
| | GPT-5-nano | 1.7 | 12.1 | 19.1 | 41.3 |
| | GPT-5 | 8.6 | 24.1 | 18.3 | 56.4 |
| Fine-tuned LLM | Qwen-2.5-7b | 3.5 | 11.9 | 18.9 | 42.9 |
| Naive ICL | Qwen-2.5-7b | 5.2 | 10.3 | 15.9 | 40.5 |
| MIPRO | Qwen-2.5-7b | 8.6 | 12.1 | 15.1 | 41.3 |
| *CORRECT* | Qwen-2.5-7b | 12.1 (+8.6) | 15.5 (+6.9) | 19.8 (+0.7) | 46.8 (+3.9) |
| | Gemini-2.5-flash | 10.3 (+5.1) | 20.7 (+6.9) | **38.9 (+7.1)** | 55.2 (-0.8) |
| | Gemini-2.5-pro | 6.9 (+1.7) | 17.2 (+5.5) | 24.6 (-0.8) | 52.4 (+2.4) |
| | GPT-5-nano | 6.9 (+5.2) | 17.2 (+5.0) | 24.6 (+5.5) | 44.4 (+3.1) |
| | GPT-5 | **17.2 (+8.6)** | **32.3 (+8.2)** | 38.1 (+19.8) | **58.8 (+2.4)** |

*Table 1.* *CORRECT* achieves higher error recognition accuracy over existing advances (WHO&WHEN dataset). Acc@0 denotes exact-step accuracy (the model must pinpoint the precise error step). Acc@k denotes tolerant accuracy (a prediction is correct if it falls within $\pm k$ steps of the ground truth). For rows corresponding to *CORRECT*, "(+X)" implies its relative improvements over LLM as a judge.

diverse tasks including multi-hop QA (HotpotQA (Yang et al., 2018), Musique (Trivedi et al., 2022), WikiMQA (Ho et al., 2020)), scientific reasoning (ARC (Clark et al., 2018), MMLU-Pro (Wang et al., 2024)), mathematical reasoning (Math500 (Lightman et al., 2023)), and general agentic tasks including planning (GAIA (Mialon et al., 2023)). WHO&WHEN consists of two subsets: a Human-Crafted subset and an Algorithm-Generated subset. Experiments are conducted on both open- and closed-source models, including the Qwen (Yang et al., 2024; 2025), Llama (Dubey et al., 2024), GPT (Hurst et al., 2024; OpenAI, 2025), DeepSeek-R1 (Guo et al., 2025), and Gemini series (Comanici et al., 2025). We mask each trajectory itself and avoid receiving its own error schema for preventing data leakage. Additional experimental details are provided in Appendix A.3, including SFT details and hyperparameters.

**Baselines.** We compare against four advances:

- *LLM-as-a-Judge*: a zero-shot prompting strategy where an LLM directly inspects trajectories without auxiliary guidance (Zhang et al., 2025; Peng et al., 2023a);
- *Fine-tuning*: Qwen-2.5-7b-Instruct is trained on the full trajectory dataset to learn domain-specific failure patterns (Chen et al., 2025; Fu et al., 2025);
- *Naive In-Context Learning*, which inserts complete error trajectories as few-shot exemplars (Yu et al., 2025).

- *MIPRO*: optimizes prompts via Bayesian search over instruction–example combinations using success signals from prior runs (Opsahl-Ong et al., 2024).

**Metrics.** Following existing advances (Zhang et al., 2025), we report *step-level accuracy*, which provides actionable debugging signals. To account for the ambiguity of error attribution, we additionally report accuracy@k, where predictions within $k$ steps of the ground truth are treated as correct, e.g., Acc@0 requires identifying the exact erroneous step, while Acc@1 tolerates an offset of one step. This better reflects practical debugging scenarios, where approximate localization is often sufficient.

We report median performance over five independent runs.

### 5.1. End-to-End Performance

**CORRECT achieves significant gains in error recognition accuracy (WHO&WHEN dataset).** Table 1 shows that *CORRECT* consistently surpasses existing advances across both human-crafted and algorithm-generated subsets. On human-crafted data, *CORRECT* raises Qwen-2.5-7B's exact-step accuracy from 3.5% to 12.1% (a 3.5× improvement), and improves GPT-5 from 8.6% to 17.2%. These gains extend to tolerant metrics as well, with GPT-5 + *CORRECT* achieving 32.3% at Acc@1 versus 24.1% for the baseline. By contrast, fine-tuning (3.5%) and naive ICL (5.2%) offer only marginal improvements, suggesting that standard

| Method | Tolerance | Dataset | | | | | | | Avg. |
|---|---|---|---|---|---|---|---|---|---|
| | | Gaia | HotpotQA | Musique | WikiMQA | Arc | Math500 | MMLU-Pro | Improv. |
| Synthesized by **GPT-4o-mini** | | | | | | | | | |
| LLM-as-a-Judge | Acc@1 | 28.6 | 34.8 | 27.8 | 14.7 | 64.0 | 10.2 | 58.3 | - |
| | Acc@3 | 42.9 | 59.4 | 77.8 | 55.9 | 75.0 | 23.4 | 62.5 | - |
| | Acc@5 | 50.0 | 63.8 | 77.8 | 64.7 | 78.0 | 35.6 | 66.7 | - |
| CORRECT | Acc@1 | 28.6 | **60.9** | **38.9** | **44.1** | **80.4** | **57.1** | **69.1** | +20.1 |
| | Acc@3 | **50.0** | **94.2** | **88.9** | **88.2** | **88.2** | **87.8** | **88.2** | +27.6 |
| | Acc@5 | **64.3** | **95.7** | **88.9** | **94.1** | **91.2** | **95.9** | **94.1** | +28.7 |
| Synthesized by **GPT-5-Nano** | | | | | | | | | |
| Baseline | Acc@1 | 16.7 | 14.7 | 11.9 | 6.44 | 62.8 | 41.8 | 50.0 | - |
| | Acc@3 | 27.8 | 48.9 | 43.9 | 35.6 | 69.6 | 57.1 | 64.7 | - |
| | Acc@5 | 38.9 | 65.8 | 58.8 | 56.1 | 71.1 | 64.3 | 70.59 | - |
| CORRECT | Acc@1 | **30.6** | **35.8** | **32.7** | **16.9** | **80.4** | **57.1** | **69.1** | +16.8 |
| | Acc@3 | **44.4** | **72.7** | **61.2** | **49.9** | **88.2** | **87.8** | **88.2** | +20.1 |
| | Acc@5 | **52.8** | **84.3** | **77.2** | **69.0** | **91.2** | **95.9** | **94.1** | +18.9 |

*Table 2.* Performance comparison across multiple datasets. All numbers report the error recognition accuracy.

supervised learning and raw in-context trajectories fail to capture complex error patterns. On algorithm-generated data, *CORRECT* maintains clear advantages, with Gemini-2.5-Flash improving from 31.8% to 38.9% (+7.1 points) and GPT-5 exhibiting the largest gain (+19.8 points). These consistent improvements across model families (Qwen, Gemini, GPT) and scales (7B to GPT-5) highlight the effectiveness of condensed error schemas.

**CORRECT delivers 17–28% average improvements (CORRECT-ERROR benchmark).** Table 2 highlights *CORRECT*'s strong generalization across seven datasets. For GPT-4o-mini subset, *CORRECT* improves average accuracy by 20.1%, 27.6%, and 28.7% at Acc@1, Acc@3, and Acc@5, respectively using Qwen-2.5-7b. Gains are especially pronounced on knowledge-intensive tasks: HotpotQA (+26.1 points), WikiMQA (+29.4 points), and Math500 (+46.9 points). At higher tolerances, performance gaps widen further: *CORRECT* reaches 94.2%, 91.2%, and 95.9% at Acc@5, compared to baseline scores of 63.8%, 78.0%, and 35.6%. *CORRECT* exhibits a similar trend in GPT-5-nano subset, with average improvements of 16.8%, 20.1%, and 18.9% across tolerance levels using Qwen-2.5-7b. Even on the challenging GAIA benchmark, *CORRECT* achieves superior scores at Acc@5.

**Strong Schema Transferability across Datasets and Models.** Figure 4a shows that schemas distilled from human-crafted trajectories transfer effectively to algorithm-generated data. Across GPT-5-nano, Gemini-2.5-Flash, and Qwen-7B, *CORRECT* with transferred schemas consistently outperforms baselines, with Gemini-2.5-Flash improving 31.8–36.5%. This cross-domain transferability indicates that distilled schemas capture fundamental error patterns.

Moreover, Figure 4b demonstrates that *CORRECT* benefits directly from model upgrades: using GPT-5 instead of Qwen-72B as the schema generator raises detection accuracy from 8.6% to 10.3% for Gemini-2.5-Flash and from 10.3% to 12.2% for Qwen-2.5-7B. These adaptive gains

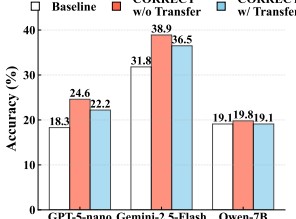
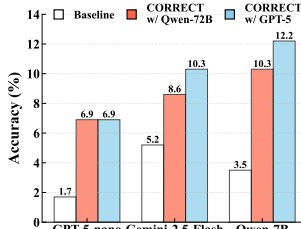

*(a)* *CORRECT* delivers improvements on Algorithm-Generated, with schemata from Hand-Crafted dataset.

*(b) CORRECT* can adaptively upgrade its performance on the Hand-Crafted dataset with model upgrade.

*Figure 4.* Ablation studies on transfer and model upgrade.

confirm that better models yield higher-quality schemas that immediately enhance downstream performance.

***CORRECT* improves performance in online deployments.** Real deployments often face continuously evolving request distributions. To evaluate this setting, we initialize CORRECT using schemata distilled from WHO&WHEN and gradually augment the cache with schemata derived from incoming HotpotQA and WikiMQA trajectories. As shown in Table 3, *CORRECT* maintains stable accuracy even when only 10–20% of the newly observed schemata are incorporated. This reflects both the strong transferability of our schemata and the small amount of on-task data required for CORRECT to adapt effectively in online environments.

| Dataset | 0–10% | 0–20% | 0–50% | 0–100% |
|---|---|---|---|---|
| Hotpot | 35.8 | 40.6 | 41.2 | 39.4 |
| WikiMQA | 70.8 | 68.6 | 67.6 | 69.1 |

*Table 3.* Streaming-style adaptation: accuracy as more schemata from new domains arrive.

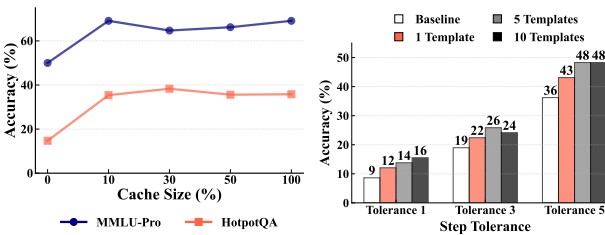

*(a) CORRECT* delivers robust improvements under different cache sizes.

*(b)* Performance of *CORRECT* with different number of error schemata on Handcrafted subset of WHO&WHEN.

*Figure 5.* Ablation studies on cache size and error schemata.

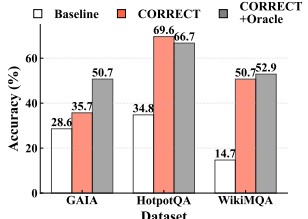

*(a)* Performance of *CORRECT* and *CORRECT* with the variation using oracle error schema.

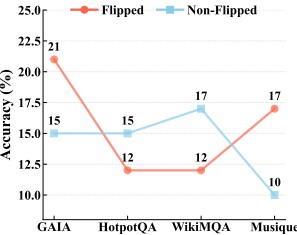

*(b)* LLMs have low performance in recognizing the errors when they encounter them.

*Figure 6.* Studies on oracle guidance and error recognition.

## 5.2. Ablation Studies

**Impact of Schema Repository Size.** Figure 5a shows *CORRECT*'s robustness to cold-start scenarios and varying cache sizes. On CORRECT-ERROR, even with only 10% of the schema library, *CORRECT* achieves 69.1% and 35.4% Acc@1 on MMLU_pro and HotpotQA, substantially outperforming baselines. Performance improves steadily with larger caches but plateaus beyond 50% (tens of schemas), suggesting that a relatively small set of diverse schemas captures most common error patterns. This logarithmic growth pattern validates our clustering-based extraction strategy.

**Impact of Number of Schemas in Online Error Recognition.** Figure 5b demonstrates that retrieving and using a single schema already greatly improves baseline Acc@1 (12.1% vs. 8.6%). Accuracy increases as more schemas are added (13.8% with 5 schemas, 15.5% with 10), though gains diminish. At higher tolerances (Acc@5), settings converge to ∼48%. These results show that a small set of well-matched schemas already efficiently captures most critical patterns, while additional schemas offer limited value.

**Comparison with Oracle Error Schema.** We compare *CORRECT* against an oracle configuration where each trajectory uses its own ground-truth schema. As shown in Figure 6a, the oracle achieves superior step-level accuracy, while *CORRECT* with 5 retrieved schemas reaches over 71.5% of oracle performance. This narrow gap shows that our semantic retrieval strategy effectively identifies schemas encoding near-equivalent knowledge to trajectory-specific patterns, validating that diverse MAS error patterns often share structural regularities.

**LLMs can hardly Recognize Their Errors During Execution.** Figure 6b shows that LLMs have limited metacognitive ability to detect their own errors during execution. When asked to identify injected errors at the exact step, they achieve only 21% accuracy on *flipped* trajectories (where errors alter the final answer) and 17–18% on *non-*

*flipped* trajectories (where errors do not affect the outcome). This reveals a fundamental limitation: *agents lack the self-awareness to recognize their own mistakes, regardless of downstream task success.* This motivates the need for external error-detection mechanisms like *CORRECT*.

***CORRECT* Introduces Negligible Overhead.** Our GPT-5 evaluations show that API usage increases only slightly ($0.86 → $0.89) on WHO&WHEN Hand-Crafted, an overhead of less than 3.5% in latency as well, because of the lightweight schema. Since schema retrieval adds only a small amount of contextual guidance to the prompt and requires no fine-tuning, the approach remains lightweight and compatible with real-time MAS deployments.

**Sensitivity to Similarity and Hotness Thresholds.** CORRECT is robust to its two primary hyperparameters: the similarity threshold $\delta$ and the schema "hotness" threshold $\theta_{\text{hot}}$. As shown in Table 5 and Table 6, varying $\theta_{\text{hot}}$ (which governs how frequently high-access schemata are refined) produces only small but consistent gains on ARC (80.4→81.3 as $\theta_{\text{hot}}$ increases from 0 to 0.3), while adjusting $\delta$ mainly trades off cache size against accuracy (76.5, 78.1, and 78.9 for $\delta=0.6, 0.7, 0.8$ respectively). They indicate that $\delta=0.7$ and refining the top 20% most-accessed schemata provide a strong balance, and that CORRECT's performance is stable.

**Decisive Interventions.** We further evaluate whether error localization can directly support targeted intervention to improve the fault tolerance of multi-agent systems. We compare targeted restart from the steps identified by *CORRECT* and from random steps on WHO&WHEN Hand-Crafted. We notice the end-to-end success rate improves from 10% to 20% with our method. This shows that even when exact Acc@0 is far from saturated, the predicted error step is still useful for downstream MAS troubleshooting and recovery.

**Robustness to Retrieval Noise.** To evaluate *CORRECT*'s robustness to retrieval noise, we inject irrelevant "hard-negative" schemata into the top-$k$ retrieval pool while keep-

ing $k=5$ fixed. Accuracy decreases moderately as noise increases (e.g., Qwen-7B drops from 12.3% to 6.9% when all five retrieved schemata are random, yet *CORRECT* consistently outperforms the baseline (3.5%). GPT-5 shows a similar trend: accuracy drops from 17.2% with clean retrieval to 12.1% when all five schemata are random, while remaining above the baseline of 8.6%. These results show that *CORRECT* does not rely on perfect retrieval. Schema guidance remains useful even when multiple partially mismatched schemata are included. Full results are available in Appendix A.14.

**Cross-Domain Versatility.** Beyond CORRECT-ERROR, we evaluate *CORRECT*'s transferability across domains, supervision formats, and benchmarks (Appendix A.13). In the error-category detection setting of Cemri et al. (2025), *CORRECT* improves recall from 15.7% to 16.6% on ProgramDev+ChatDev and from 11.9% to 15.8% on AG2+GSM8K, despite using no step-level annotations (Table 7). On AgentErrorBench (Zhu et al., 2025), *CORRECT* also improves GPT-5-nano tolerant accuracy on GAIA (36% to 46%), ALFWorld (22% to 35%), and WebShop (16% to 24%) (Table 8). These results show that the distilled schemata transfer across datasets, task formulations, and agentic environments.

## 6. Limitations

We acknowledge the following limitations. First, decisive error labels may be inherently subjective, even in human-crafted or human-annotated trajectories, since failures in MAS often involve multiple interacting agents and cascading effects. This ambiguity can introduce instability for future training, evaluation, or online adaptation that relies on such labels. Second, while *CORRECT* shows that reusable error schemata are effective for error recognition, how to better integrate these schemata with skill systems in multi-agent frameworks remains an important direction. Finally, *CORRECT* currently retrieves schemata mainly through semantic similarity, and it remains unclear whether this is the best retrieval signal, despite its robustness to retrieval noises and low costs. We leave the exploration of more structured retrieval mechanisms, potentially with a dedicated retrieval agent that reasons over schemata, agent roles, tool states, and causal dependencies, to future work.

## 7. Conclusion

We introduced *CORRECT*, the first schema-guided framework that distills recurrent MAS failures into compact, reusable schemata, enabling accurate, lightweight, and training-free identification of decisive errors in new runs. Complementing this, we release CORRECT-ERROR, a large-scale, high-fidelity benchmark capturing realistic error

patterns. *CORRECT* significantly improves error recognition accuracy, offering a practical, generalizable path toward reliable, interpretable, and scalable MAS deployment.

## Acknowledgements

We thank the anonymous reviewers for their constructive and insightful feedback. This work was supported in part by an Amazon Research Award and awards from NVIDIA Academic Program and Gemini Academic Program. It also utilized the Delta system at the National Center for Supercomputing Applications (NCSA) through allocation CIS240236 from the ACCESS program.

## Impact Statement

This paper presents work whose goal is to advance the reliability and interpretability of multi-agent systems by enabling precise and efficient error recognition. As multi-agent systems are increasingly deployed in real-world settings, such as software engineering, scientific discovery, and automated decision, improving their debuggability can lead to safer, more robust, and more maintainable AI systems. By identifying decisive errors and their propagation paths, our approach will help practitioners prevent cascading failures, reduce operational costs, and support responsible system upgrades and monitoring.

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

# A. Appendix

## A.1. Impact Statement

**Impact Statement.** This paper presents a method for improving error recognition in large language model–based multi-agent systems, with the goal of enhancing their reliability, interpretability, and maintainability. By enabling more accurate and efficient diagnosis of failure sources without additional training, our work can reduce debugging costs and support safer deployment of complex AI systems in research and real-world applications. We do not foresee significant negative societal impacts arising from this work when used responsibly.

## A.2. MAS failure trajectories are long and complex

We show the distribution of MAS trajectories as in 7. About 17.2% of the trajectories exceed the length limit of max content limit of Qwen models.

## A.3. Specifics of experimental settings

We implement our evaluation pipeline based on Zhang et al. (2025). We host open-source models using vLLM(Kwon et al., 2023) and access GPT-series models(Achiam et al., 2023) via the OpenAI API. To handle long contexts exceeding standard model limits for Qwen models(Yang et al., 2025), we employ RoPE(Su et al., 2024) scaling with 4× length extension using the "yarn"(Peng et al., 2023b) scaling type. To simulate realistic deployment scenarios where ground truth is unknown, we exclude the correct answer from evaluation prompts. For our method, we first generate all the error schemata using GPT-5 model. We then derive a similarity mapping to assign error schemata based on the semantic embedding decoded by BAAI-BPE-M3 model(Chen et al., 2024).

As shown in Appendix A.10, *CORRECT* is robust to aggressive truncation: using 500, 1000, or 2000 characters per turn yields nearly identical performance, indicating that long MAS logs (often 60k+ tokens) do not pose a practical limitation for schema retrieval.

To avoid the data leakage, we mask each trajectory itself and avoid receiving its own error schema. We decide the number of error schemata from the experiments using Qwen-2.5-7b models on Hand-Crafted dataset, Algorithm-Generated dataset, and HotpotQA dataset of CORRECT-Error. We use the same number of error schemata across all models and all datasets in CORRECT-Error. Specifically, we use 1 error schema for all experiments on the Algorithm-Generated dataset, 10 error schemata for all experiments on the Hand-Crafted dataset, and 5 error schemata for all experiments on CORRECT-Error.

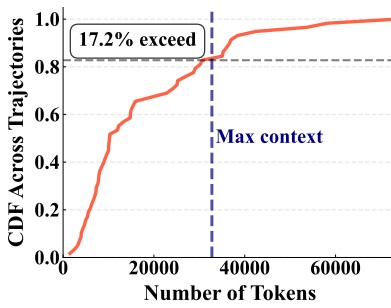

*Figure 7.* MAS failure traces are complex and long, often exceeding the model capacity.

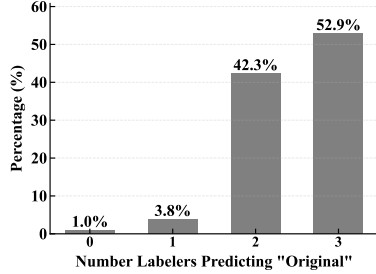

*Figure 8.* Percentage of human labelers to believe the trajectory is not synthesized.

**Fine-Tuning Baseline Details.** Following prior work (Zheng et al., 2024; Peng et al., 2023a), we train the model with a standard cross-entropy loss over the assistant responses. To avoid data leakage, SFT for the Hand-Crafted evaluation uses training data from the Algorithm-Generated split, and vice versa.

Each training instance is formatted as:

```
messages = [
{"role": "system", "content": "You are an AI assistant specialized in analyzing
multi-agent conversations to identify errors."},
{"role": "user", "content": prompt},
{"role": "assistant", "content": answer}
]
```

Only the assistant output is used as the training target.

We perform a grid search over learning rates {1e-6, 5e-6, 1e-5, 5e-5} and batch sizes {8, 16, 32}. The selected hyperparam-

*Figure 9.* Error Schema Generation

---

**Error Schema Generation**

"""Given an error analysis from a multi-agent conversation, create an error schema to help identify similar errors in the future.
Context:
Question: {question}
Ground Truth: {ground_truth}
Error Agent: {mistake_agent}
Error Step: {mistake_step}
Error Reason: {mistake_reason}
Conversation History: {chat_content}
Based on this error case, please create a error schema that will help IDENTIFY similar errors in future conversations. Focus primarily on recognition patterns rather than mitigation strategies. The schema should include:
1. Error Signatures: - What distinctive patterns or signals indicate this type of error is occurring? - What are the telltale signs in the agent's behavior or responses?
2. Error Context Analysis: - What contextual conditions typically surround this type of error? - What sequence of interactions tends to precede this error?
3. Detection Heuristics: - What specific questions can be asked to determine if this error is present? - What analytical framework can help identify this error pattern? - What key phrases or conversation patterns serve as reliable indicators?
Please format your response as a structured schema that focuses specifically on ERROR IDENTIFICATION, not on how to improve agent behavior.
Provide a concise, actionable schema in the following format:
Agent Name: {mistake_agent}
Step Number: {mistake_step}
Reason for Mistake: [Your analysis of why this specific error occurred and how to identify similar patterns] """

---

eters for the two SFT models are:

|  | Learning Rate | Batch Size |
|---|---|---|
| Hand-Crafted SFT | 5e-5 | 16 |
| Algorithm-Generated SFT | 1e-5 | 32 |

These models are trained on the full trajectory dataset for each split to encode domain-specific failure patterns.

### A.4. More details of the CORRECT-Error

We implemented a variant of Magentic-One (Fourney et al., 2024) using selector group based workflow control using AutoGen (Wu et al., 2024) to generate CORRECT-Error. Apart from the figures we showed in section 4, we observed strong inter-annotator consensus: 94.4% of synthetic trajectories fooled at least two labelers, while 52.9% were unanimously mistaken for genuine errors. We show the distribution in Figure 8

### A.5. Prompts for Offline Error Schema Generation

We show the prompts we used for offline schema generation in Fig 9.

### A.6. Prompts for online schema-guided generation

We show the prompts we used for online schema-guided generation in Fig 10.

### A.7. Comparison to Human-Curated Taxonomies

We further compare and contrast the error schemata generated by CORRECT with human-curated taxonomies in MAST:

*Figure 10.* Schema-guided generation

| Schema-guided generation |
|---|
| "HOW TO USE THIS REFERENCE EXAMPLE:"
"This template demonstrates one type of error pattern for reference. To apply it to your analysis:"
"1. Study the ERROR PATTERN shown: What type of mistake does this example identify?"
"2. Use this as reference to analyze YOUR conversation:"
" • Read through your conversation systematically (Step 0, Step 1, Step 2...)"
" • At each step, ask: 'Is there an error here, and does it match this pattern or a different one?'"
" • The error in your case may follow the same pattern or be completely different"
"3. Remember this is just a reference example:"
" • Your error may occur at any step number"
" • Your error may be a different type entirely"
" • Use this template to help you recognize what errors look like, not to assume your error matches" |

**Error Schemata vs. Static Human Taxonomies** Human-curated taxonomies face several limitations:

- *Lack of granularity.* Human-defined categories (e.g., "wrong tool use," "missing precondition") operate at coarse conceptual levels and do not provide the fine-grained, step-specific patterns required for localizing errors within trajectories.

- *High manual cost and slow iteration.* Curating and maintaining a taxonomy requires extensive domain expertise. As models, tools, and task distributions shift, these taxonomies quickly become outdated.

- *Limited adaptability under distribution drift.* Static taxonomies cannot capture new, emerging failure modes as the environment or agents evolve.

**Advantages of CORRECT.** CORRECT automatically distills fine-grained, step-level schemata from observed trajectories and updates them asynchronously without adding online inference overhead. This enables CORRECT to track evolving failure modes and maintain accuracy under distribution drift—capabilities that static human-curated taxonomies cannot provide in large-scale deployments.

### A.8. Human Verification, Automated Validation, and Scalability

Human involvement during schema construction can introduce overhead, but CORRECT is not dependent on manual supervision. In practical deployments (e.g., conversational systems such as ChatGPT or Gemini), user feedback is already collected passively and can naturally serve as weak supervision without incurring additional annotation costs. Moreover, each schema is reused across many queries, so even a small number of validated schemata provides substantial amortized benefit: as shown in Figure 8, only 58 schemata are sufficient to improve HotpotQA detection accuracy from 14.7 to 35.4.

When human input is unavailable, CORRECT can instead employ LLM-as-a-judge with confidence filtering to automatically validate successful trajectories before inserting new schemata into the cache. This aligns with recent efforts such as EcoAssistant, which leverage human-verified or automatically filtered caches to improve long-horizon reasoning. Exploring more automated or semi-supervised schema-validation pipelines—while preserving interpretability and reliability—is a promising direction for future work.

### A.9. Prompt for Natural Error Injection

### A.10. Robustness to Context-Length Truncation

We use the BGE-M3 encoder (context window 8192) to embed multi-agent trajectories. Because raw MAS logs can exceed 60k–70k tokens, we truncate each turn to a fixed character budget before embedding. Table 4 shows that CORRECT is highly robust to this truncation: using 500, 1000, or 2000 characters per turn yields nearly identical performance.

These results show that CORRECT's retrieval mechanism remains stable even under aggressive truncation, and long-context MAS logs do not hinder schema matching in practice.

---

**Error injection**

```
You are analyzing an error pattern to create an injection plan.

THOUGHT TEMPLATE (Error Detection Guidelines):
{thought_template}

ERROR INFORMATION:
- Agent:  {mistake_agent}
- Step:  {mistake_step}
- Reason:  {mistake_reason}

ERROR TRAJECTORY:
{error_formatted}

CORRECT TRAJECTORY:
{correct_formatted}

CRITICAL REQUIREMENTS:
1.  Injection point must be semantically similar to the original error.
2.  Error must flow naturally from that point.
3.  Consider cascade effects on all later steps.
4.  For consecutive CodeExecutor steps, modify only the final step's output.

STEP-BY-STEP ANALYSIS REQUIRED:
Step 1:  Analyze the original error and identify what went wrong.
Step 2:  Map to semantically similar steps in the correct trajectory.
Step 3:  Select the proper injection point.
Step 4:  Describe how the error should be replicated.
Step 5:  Assess how the injected error propagates.

VERIFICATION CHECKLIST:
- Semantically similar injection point?
- Natural error flow?
- Cascade effects considered?
- Final CodeExecutor step modified if consecutive?
- Realistic and agent-consistent error?

Output format:
<injection_step>[...]</injection_step>
<injection_agent>[...]</injection_agent>
<error_pattern>[...]</error_pattern>
<injection_strategy>[...]</injection_strategy>
<expected_impact>[...]</expected_impact>
```

```
You are tasked with modifying a message to inject a specific error.
Follow the steps below carefully.

ORIGINAL MESSAGE:
Agent: {agent}
Content: {original_content}

INJECTION PLAN:
Error Pattern: {injection_plan['error_pattern']}
Strategy: {injection_plan['error_strategy']}
Expected Impact: {injection_plan['expected_impact']}

INSTRUCTIONS: Complete each section below in order.  Show your thinking
process.

<STEP1_ANALYSIS>
- What is the agent's formatting style?
- What was the agent trying to communicate?
- What is the context of this message?
</STEP1_ANALYSIS>

<STEP2_ERROR_UNDERSTANDING>
- What specific error am I injecting?  {injection_plan['error_strategy']}
- Why would this error naturally occur?
- How does this relate to the error pattern {injection_plan['error_pattern']}?
</STEP2_ERROR_UNDERSTANDING>

<STEP3_MODIFICATION_PLAN>
- What exact change will I make?
- How will I maintain the agent's style?
- Why is this change realistic?
</STEP3_MODIFICATION_PLAN>

<STEP4_VERIFICATION>
☐ Maintains {agent}'s style and format?
☐ Implements strategy exactly?
☐ Believable to the agent?
☐ Causes {injection_plan['expected_impact']}?
☐ Within agent capabilities?
☐ Realistic error?
☐ Leads to incorrect final answer?
</STEP4_VERIFICATION>

<MODIFIED_CONTENT>
[Put ONLY the modified content here, exactly as the agent would output it]
</MODIFIED_CONTENT>
```

*Table 4.* Effect of per-turn truncation length on schema retrieval accuracy.

| Dataset | 500 chars | 1000 chars | 2000 chars |
|---|---|---|---|
| Hand-Crafted | 12.2 | 12.2 | 12.2 |
| Algorithm-Generated | 19.8 | 19.8 | 19.2 |

## A.11. Failure Cases: Multi-Error Trajectories

CORRECT may misfire when a trajectory contains multiple errors and a later, higher-salience failure dominates the schema match. Because CORRECT follows the decisiveness definition of (Zhang et al., 2025), it is designed to identify the earliest decisive error; however, when downstream errors exhibit stronger structural signals, the retrieved schema may align more closely with these later steps.

**Example.** In the example below, the human-labeled decisive error occurs at Step 4:

> *"WebSurfer failed to locate the specific volume in the University of Leicester paper due to incomplete data retrieval and insufficient PDF analysis."*

CORRECT instead predicts Step 12 as the decisive error:

> *"The agent remained on the search-results page instead of navigating into the DOI page containing the required endnote information."*

**Schema Match.** The retrieved schema (abridged) emphasizes:

- **Error signature:** incomplete or insufficient search criteria,
- **Context:** tasks requiring precise filtering of external data sources,
- **Heuristic:** "Is the agent's search action complete and accurate?"

Both Step 4 and Step 12 satisfy these conditions, but Step 12 provides a more overt instance of incorrect search behavior, making it a stronger surface-level match for the schema.

**Discussion.** Such multi-error trajectories represent the primary category where CORRECT may deviate from human-annotated earliest-error labels. In practice, they are uncommon, but they highlight an inherent challenge of schema-guided detection when structurally similar failures occur at multiple points in a trajectory.

## A.12. Sensitivity to similarity and hotness threshold

*Table 5.* Sensitivity to the schema-refinement threshold $\theta_{\text{hot}}$.

| $\theta_{\text{hot}}$ | 0.0 | 0.1 | 0.2 | 0.3 |
|---|---|---|---|---|
| ARC Accuracy | 80.4 | 80.6 | 81.2 | 81.3 |

**Effect of $\delta$.** Lowering the similarity threshold reduces cache size but slightly harms accuracy, whereas $\delta = 0.7$ offers a strong balance:

*Table 6.* Sensitivity to the similarity threshold $\delta$.

| $\delta$ | 0.6 | 0.7 | 0.8 |
|---|---|---|---|
| ARC Accuracy | 76.5 | 78.1 | 78.9 |

## A.13. Transferability to Alternative Domains and Supervision Formats

We further evaluate CORRECT's cross-domain and cross-formulation generalization under settings that differ substantially from our primary step-level error localization task.

**Transfer to Error-Category Detection.** The dataset introduced in (Cemri et al., 2025) provides error *categories* but does not include step-level annotations. This differs from our formulation, where CORRECT predicts the earliest decisive step in a trajectory. Because of this structural mismatch, a direct step-level comparison is not possible. To evaluate cross-formulation

*Table 7.* CORRECT adapted to the error-category detection setting of (Cemri et al., 2025).

| Dataset | Baseline | CORRECT |
|---|---|---|
| ProgramDev + ChatDev | 15.7 | 16.6 |
| AG2 + GSM8K | 11.9 | 15.8 |

transferability, we adapt CORRECT by providing only the retrieved schemata and prompting the model to output an error *type* instead of a step index. Using Qwen-7B, CORRECT improves recall on both released datasets:

Despite lacking step-level supervision, CORRECT retains consistent gains, indicating that its schemata encode transferable semantic error patterns that remain effective under coarser labeling regimes.

**Transfer to AgentErrorBench.** We further evaluate CORRECT on AgentErrorBench, which contains heterogeneous agentic environments spanning embodied navigation, web interaction, and long-horizon planning. This benchmark differs substantially from CORRECT-Error in both task structure and failure modes. We conduct experiments using GPT-5-nano and report tolerant accuracy within $\pm 1$ step. Results are summarized in Table 8.

*Table 8.* Cross-domain transfer results on AgentErrorBench using GPT-5-nano.

| Dataset | Baseline | CORRECT | Improvement |
|---|---|---|---|
| GAIA | 36% | 46% | +10% |
| ALFWorld | 22% | 35% | +13% |
| WebShop | 16% | 24% | +8% |

CORRECT achieves consistent improvements across all evaluated environments, with the largest gains observed on ALFWorld, which requires long-horizon planning and precise tool usage. These results demonstrate that CORRECT's distilled schemata generalize effectively to previously unseen domains and interaction modalities.

## A.14. Robustness to Retrieval Noise

To assess CORRECT's resilience to mismatched schema retrieval, we inject random, irrelevant schemata ("hard negatives") into the retrieval pool while keeping the total number of retrieved schemata fixed at $k=5$. The tables below report performance as the number of injected random schemata increases.

*Table 9.* Impact of random (irrelevant) schemata injected into the retrieval pool (Qwen-7B).

| Model | Baseline | CORRECT | +1 Rand | +3 Rand | +4 Rand | +5 Rand |
|---|---|---|---|---|---|---|
| Qwen-7B | 3.5 | 12.3 | 10.7 | 10.7 | 10.7 | 6.9 |

*Table 10.* Impact of retrieval noise on GPT-5.

| Model | Baseline | CORRECT | +1 Rand | +3 Rand | +4 Rand | +5 Rand |
|---|---|---|---|---|---|---|
| GPT-5 | 8.6 | 17.2 | 13.8 | 15.5 | 13.8 | 12.1 |

Even with multiple hard negatives included, CORRECT consistently improves over the baseline, demonstrating strong robustness to retrieval noise and confirming that schema-guided reasoning does not depend on perfect semantic matching.

