# OpenReview forum: "CORRECT: COndensed eRror RECognition via knowledge Transfer in multi-agent systems"
_ICML.cc/2026/Conference — ICML 2026 regular_

### Official Review · Reviewer_s1TG · 2026-03-11

**Soundness:** 3
**Presentation:** 3
**Significance:** 3
**Originality:** 3
**Overall Recommendation:** 5
**Confidence:** 3

**Summary:**

This paper solves the decisive error recognition problem in multi-agent systems (MAS), where the goal is to find the earliest misstep that, if corrected, would flip a failed trajectory to success. It introduces CORRECT, a training-free framework that distills past failure trajectories into compact error schemata, stores them, and retrieves relevant ones at inference time to guide an LLM in diagnosing new failures. Processing happens in three phases: offline schema extraction from clustered failures, online schema-guided recognition via semantic retrieval, and dynamic schema management through expansion and distillation. A companion benchmark which the authors introduce, CORRECT-Error, provides 2,000+ annotated MAS trajectories across seven tasks, built using human feedback. Results show up to sizeable improvement on the WHO&WHEN dataset and CORRECT-Error over LLM-as-a-judge baselines.

**Compliance With Llm Reviewing Policy:**

Affirmed.

**Final Justification:**

The author's review rebuttals and replies to my questions are helpful and addressed the main concerns and weaknesses I had with the paper. I raise my score to a 5.

**Key Questions For Authors:**

1. The best Acc@0 sits around 17% on hand-crafted examples. Does this level actually assist real-world MAS troubleshooting? Why is there such a large gap between algorithm-generated and hand-craft examples?

2. Since CORRECT-Error injects exactly one error, the decisive error always matches the injection point. Real MAS failures may have multiple contributing errors where the "earliest decisive" one is ambiguous. How does CORRECT handle multi-error trajectories (mentioned briefly in Appendix but deserving more attention)?

3. What happens when the schema cache encounters a truly novel error with no close match? Does CORRECT degrade gracefully to baseline performance, or does a mismatched schema actively hurt?

**Limitations:**

Yes

**Strengths And Weaknesses:**

Strengths:
- Soundness: Thorough experiment results across several LLM backends, two datasets (WHO&WHEN, CORRECT-Error), and multiple baselines. Ablation studies probes cache size, schema count, similarity thresholds. Cross-domain transfer (Figure 4a) and streaming adaptation (Table 3) results show practical robustness.
- Presentation: Paper is well organized and clearly laid out into logical sections. Strong motivation section that examines past methods and makes their gaps visible. Methodology section is easy to follow with Algorithm 1 providing a clean step-by-step view.
- Significance: The problem being solved is relevant to real-world challenges faced by LLM agents. Tracing chain reactions between agents is especially difficult to debug. A key insight, that many failure sequences look alike in meaning, is backed by data and motivates the entire approach. The training-free, cache-based design fits changing environments where error distributions shift over time.
- Originality: Shifting from raw trajectory data to compressed schematized patterns avoids clutter while boosting relevance. The schema expansion and distillation mechanisms for online adaptation are thoughtful engineering choices targeting real-world usability.

Weaknesses:

- Soundness: Justification for the prompts used and more details regarding CORRECT hyperparameters would be nice. Testing of CORRECT on a dataset generated by same authors (CORRECT-Error) has potential for bias.
- Presentation: An architectural diagram showing the overall approach is missing. It would be very useful to be able to visualize how each component of CORRECT relate to each other. Some of the figures are too small to properly see without zooming in (Figure 3).
- Significance: Absolute performance remains low. Best Acc@0 is 17.2% with GPT-5 on the hand-crafted set, while Acc@1 reaches only 37.8%. The paper should explore in more depth whether 17% accuracy is actionable for the stated goal of enabling targeted interventions.
- Originality: At its core, CORRECT follows a familiar retrieval augmented generation (RAG) workflow (querying cache to retrieve schemata for LLM context). The application domain is new, but the technical mechanism is standard RAG. More in depth related work and acknowledgement about this foundation would strengthen the work.

---

> ### Author Rebuttal · Authors · 2026-03-31
>
> **We sincerely appreciate your valuable feedbacks.**
>
> **W2 Presentation**
> Thank you for the suggestion! We will incorporate this feedback in the camera-ready version by adding a clearer end-to-end architectural diagram and enlarging the existing figures for better readability
>
> **W4 Originality**
> Thank you for this suggestion. We agree that CORRECT is related to the broader **RAG** paradigm at a high level. However, our contribution is not retrieval itself, but a new **formulation and system design for multi-agent error diagnosis**.
>
> Specifically, CORRECT introduces **error schemata** as reusable, structured abstractions of failure patterns, and formulates decisive error localization as a retrieval-and-instantiation problem over these schemata. Unlike standard RAG, which retrieves static knowledge or documents, CORRECT retrieves **procedural diagnostic heuristics** for reasoning about failure causes. This distinction is also supported empirically: in Table 1, **naive ICL achieves 5.2 Acc@0 on WHO&WHEN, while CORRECT reaches 12.0 Acc@0**, suggesting that the key contribution lies in the schemata abstraction rather than retrieval alone.
>
> CORRECT also includes **cache expansion** and **schemata distillation** mechanisms, allowing the repository to evolve over time, and we construct **CORRECT-Error**, a dataset that captures decisive failure steps through controlled error injection and human validation. Together, these design choices make CORRECT fundamentally different from standard RAG pipelines built over static corpora.
>
> We will strengthen the related work discussion in the final version to better position CORRECT with respect to RAG and clarify these distinctions.
>
> **W1, W3, Q1: Soundness and Significance**
>
> Thank you for this question. We believe the current accuracy is already practically useful because decisive error localization can directly support targeted intervention. In a follow-up experiment on the hand-crafted trajectories, we restart execution from the identified decisive step using the corrected target step, which improves the end-to-end success rate from **10% to 20%**. This shows that even when exact **Acc@0** is far from saturated, the predicted error step is still useful for downstream MAS troubleshooting and recovery. We will include this result in the camera-ready version.
>
> We also observe gains on **real-world human-labeled data**. On the **MAST** dataset (Appendix A.13), CORRECT improves performance from **15.7 → 16.6** on ProgramDev+ChatDev and from **11.9 → 15.8** on Ag2+Gsm8k, showing that the method provides useful diagnostic signals beyond synthetic benchmarks.
>
> Regarding the gap between algorithm-generated and hand-crafted examples in **WHO&WHEN**, the two subsets come from different MAS sources, with the hand-crafted subset drawn from **magnetic-one** and the algorithm-generated subset from **captain agent**. This makes the hand-crafted setting inherently harder, which is consistent with the lower absolute scores reported in the original WHO&WHEN paper.
>
> **Q2**
> Thank you for this important point. We agree that real MAS failures can involve multiple contributing errors.
>
> Our formulation follows the **decisiveness definition** used in prior work: the target is the **earliest decisive error**, i.e., the first step whose correction would change the final trajectory outcome from failure to success. Under this definition, later errors are often downstream consequences of the earliest failure. Focusing on the earliest decisive error is practically useful because it enables targeted intervention, such as restarting from that step, correcting the intermediate state, or re-planning from a clean prefix.
>
> While CORRECT-Error injects a single primary error for controlled evaluation, the resulting trajectories can still exhibit **error cascades** in which the initial mistake propagates into multiple downstream failures.
>
> Importantly, **CORRECT itself is not restricted to single-error scenarios**. The same framework can be extended to iterative diagnosis: after identifying and correcting one error, the system can re-run and detect subsequent failures. We will clarify this extension more explicitly in the final version.
>
> **Q3**
> Thank you for this question. Our results show that CORRECT does not rely on exact schemata matches and degrades gracefully when matches are imperfect.
>
> Figure 4a shows strong schemata transferability across datasets. For example, **Gemini-2.5-Flash improves by 31.8–36.5%**, suggesting that retrieved schemata capture reusable error patterns rather than brittle matches.
>
> In a hard-negative setting with random irrelevant schemata, CORRECT still outperforms the baseline (e.g., **GPT-5: 8.6 → 15.5**), indicating that mismatched schemata do not harm performance.
>
> Finally, CORRECT includes a **schemata expansion mechanism**: when no close match exists, a new schemata is added, allowing the system to adapt to novel errors over time.

---

> > ### Author Rebuttal · Reviewer_s1TG · 2026-04-03
> >
> > Thanks for the detailed rebuttal, most of my concerns have been addressed. The experiments on the effectiveness of CORRECT even with irrelevant schemata retrieved (Q3) are compelling.
> >
> > The concern about originality (W4) remains. The authors argue that CORRECT is distinct from typical RAG as it retrieves "procedural diagnostic heuristics" as opposed to documents. However, it still follows a typical process of inserting a query, retrieving similar items from a cache, and using them to populate a LLM prompt. The improvement over the baseline on WHO&WHEN (5.2 vs. 12.0) suggests that schemata play a role in performance improvement. However, it is not clear whether there is improvement from a novel retrieval process or as a result of a better-populated cache.
> >
> > Even though there is an improvement in end-to-end success rate from 10\% to 20\%, the accuracy of 17.2\% implies that the system is still choosing wrong step in more than 80\% of the cases. Given the stated goal of enabling targeted interventions, this is somewhat concerning. Furthermore, although CORRECT-Error only has a single error per trajectory, real world failures often involve multiple independent root causes. Could the authors provide evidence that CORRECT is able to handle such scenarios, rather than a single injected error setup where downstream failures result from a single source?

---

> > > ### Author Response · Authors · 2026-04-04
> > >
> > > We want to kindly note that our contribution is not a fundamentally new retrieval operator, but a **better-populated and better-maintained schema cache**, and the empirical evidence consistently supports this.
> > >
> > > First, **retrieval alone is insufficient**. On WHO&WHEN, naively retrieving the similar examples (i.e., RAG+ICL without our optimization) achieves only *5.2% Acc@0*, while CORRECT reaches *12.0%*, more than doubling performance. Since both approaches rely on retrieval, this gap indicates that the improvement comes from what is stored **(distilled schemata) rather than the retrieval process itself**.
> > >
> > > Second, the cache expansion and distillation mechanisms directly demonstrate that improving the cache leads to better performance. In the streaming setting (Table 3), **incorporating new schemata yields consistent gains (e.g., 31.8% → 36.5%), showing that simply augmenting the cache improves generalization across domains**.
> > >
> > > Most importantly, Table 5 and Table 6 provide controlled evidence that **refinement and cache quality—not retrieval—drive performance**. In Table 5, varying the refinement threshold θ_hot improves accuracy from 80.4 → 81.3, showing that selectively refining high-usage schemata leads to measurable gains. In Table 6, adjusting the similarity threshold δ (which governs cache composition) yields consistent improvements (76.5 → 78.9 as δ increases), highlighting the importance of maintaining a well-structured and properly filtered cache. *Notably, these gains are achieved without changing the retrieval mechanism, isolating the effect of cache design*.
> > >
> > > Together, these results show that CORRECT’s performance is primarily driven by **how the schema repository is constructed, expanded, and distilled, rather than by innovations in retrieval itself**.
> > >
> > > We also sincerely appreciate for raising this important concern of practical usage. We agree that the exact Acc@0 of 17.2% alone may appear low if interpreted strictly as a binary success metric. However, **decisive error localization is inherently tolerant, and exact-match accuracy significantly underestimates its practical usefulness**.
> > >
> > > When we evaluate CORRECT under a tolerance window (i.e., allowing predictions within a small distance of the ground-truth decisive step), **performance improves substantially**. For GPT-5 on the hand-crafted set, accuracy increases from *17.24% (exact match) to 37.93% / 41.38% / 46.55% / 51.72% / 55.17%* as the tolerance window expands. This indicates that a **large portion of predictions are near-misses**, which still effectively narrows down the error region.
> > >
> > > Regarding single error detection, we want to kindly clarify that our formulation follows the **standard, popular definition of decisive error** adopted in prior work [1]: the target is the **earliest intervention** point whose correction **flips** the final outcome. This definition is well-defined even in complex trajectories, and we validate it on real human-annotated data (WHO&WHEN), where errors are not synthetically injected but arise naturally from multi-agent interactions. The consistent improvements on this dataset suggest that CORRECT already operates beyond purely synthetic single-error settings.
> > >
> > > That said, we agree that *real-world failures may involve multiple independent root causes, which are underexplored in current benchmarks*. Importantly, **this limitation is not unique to CORRECT**. Moreover, once the earliest decisive error is corrected, the downstream trajectory—and thus the manifestation of later errors—can change significantly. As a result, **later error patterns observed in the original failed trajectory may no longer persist or may take a different form after intervention**, further reinforcing the importance of identifying the earliest causal failure rather than jointly modeling all downstream errors in a single pass.
> > >
> > > We did a literature survey, yet we cannot find any real dataset containing realistic multi-step error labels. We instead address this gap by evaluating on **real-world agent datasets in the Appendix A.13**, including MAST and AgentErrorBench, which may involve more complex, overlapping error patterns. On MAST, CORRECT improves recall from *15.7 to 16.6 on ProgramDev+ChatDev and from 11.9 to 15.8 on AG2+GSM8K.* On AgentErrorBench, CORRECT also shows consistent gains across heterogeneous environments using GPT-5-nano with tolerant accuracy (±1 step): *GAIA 36% → 46%, ALFWorld 22% → 35%, and WebShop 16% → 24%*. These results suggest that CORRECT **remains effective beyond the controlled single-error setup and can provide useful diagnostic signals in more realistic settings**.
> > >
> > > [1]: Zhang, Shaokun, et al. "Which agent causes task failures and when? on automated failure attribution of llm multi-agent systems." arXiv preprint arXiv:2505.00212 (2025).

---

### Official Review · Reviewer_KPnZ · 2026-03-12

**Soundness:** 4
**Presentation:** 3
**Significance:** 2
**Originality:** 3
**Overall Recommendation:** 5
**Confidence:** 4

**Summary:**

The CORRECT framework addresses the critical challenge of decisive error recognition in multi-agent systems, where minor mistakes often cascade into complex, intertwined failure trajectories. Recognizing that MAS failures frequently recur with similar structural patterns despite surface-level differences, CORRECT introduces a lightweight, training-free approach that distills historical error logs into compact, reusable “error schemata”. During online deployment, when a new failure occurs, the system retrieves the most semantically relevant schemata from its cache to guide a LLM in localizing the exact agent and step responsible for the error. This targeted, schema-guided inference avoids the high costs of model fine-tuning and the context-window limitations of naive in-context learning. To support rigorous evaluation, the authors also developed CORRECT-Error, a large-scale benchmark of over 2,000 annotated trajectories created through a realistic error-injection pipeline. Extensive experiments across diverse MAS tasks demonstrate that CORRECT improves step-level error localization accuracy by up to 19.8% compared to existing methods, while maintaining near-zero computational overhead and strong adaptability to dynamic environments.

**Compliance With Llm Reviewing Policy:**

Affirmed.

**Final Justification:**

The rebuttal has effectively addressed several of my previous concerns, particularly regarding the potential extension of CORRECT to other agent-related domains. In light of these clarifications, I am glad to raise my rating to 5.

**Key Questions For Authors:**

[Q1] Whether these localized failures can be transformed into reusable episodic memories to drive continuous policy improvement? I acknowledge that there is a non-trivial algorithmic gap between diagnostic credit assignment (error localization) and generative policy adaptation. Nevertheless, expanding the discussion on how CORRECT's “error schemata” could serve as a foundational memory mechanism for self-improving agents would substantially elevate the paper's forward-looking significance and broader impact.

[Q2] A critical dimension that warrants further investigation is CORRECT's error localization performance in long-horizon tasks. Given the extensive noise and complexity inherent in prolonged MAS interactions, what I'm curious about is these extended erroneous trajectories still exhibit the high structural similarity observed by the authors?

**Limitations:**

Please refer to the above weakness and questions.

**Strengths And Weaknesses:**

[S1] The problem identified in this work is highly significant. The authors empirically observe that over 80% of failure trajectories exhibit a semantic similarity exceeding 0.8, providing compelling evidence that error patterns in multi-agent systems are not stochastic, but rather exhibit strong structural recurrence.

[S2] To ensure the robust reusability of error schemata, the authors implement a series of meticulously designed operations. Crucially, this mechanism transcends naive text summarization; it functions as a targeted distillation process that isolates and preserves highly transferable error-detection heuristics.

[S3] Beyond proposing the CORRECT framework, the authors construct a benchmark, CORRECT-Error, to facilitate rigorous evaluation. Furthermore, they conduct a comprehensive suite of empirical analyses, which collectively substantiate the methodological soundness of the paper.

[W1] The core efficacy of the CORRECT is intrinsically tied to the reusability of error schemata, making it highly advantageous in environments with structurally recurrent failure patterns. Conversely, in highly open-ended scenarios where errors are predominantly heterogeneous and idiosyncratic, the utility of schema retrieval may naturally diminish.

[W2] Although the construction of the CORRECT-Error is highly methodical, its heavy reliance on a synthetic error-injection pipeline remains a notable limitation. The authors assert high fidelity via human validation, which is a commendable strength. However, the ultimate soundness of the empirical conclusions intrinsically hinges on whether these injected failures accurately capture the complex causal and temporal dynamics of naturally occurring decisive errors in real-world MAS deployments. Human validation, while a strong mitigating factor, cannot entirely bridge the gap between synthetic perturbation and organic failure cascades.

[W3] While this paper tackles an undoubtedly important task, its scope remains relatively narrow. Compared to foundational challenges in the field, such as generalized agent reliability, robust planning, or long-term memory, the problem addressed here is more of a localized diagnostic issue. Given this constrained scope, the broader impact of the contribution is somewhat limited, which justifies my rating of ‘Fair’ for significance.

---

> ### Author Rebuttal · Authors · 2026-03-31
>
> **We sincerely appreciate your valuable feedbacks.**
>
> **w1/w2: Dependence on recurrent error structure and limited validation beyond synthetic error-injection benchmarks**
>
> Thank you for these insightful comments. While we agree that CORRECT is most naturally suited to settings with some structural recurrence, our empirical results suggest that such patterns are prevalent across a wide range of MAS tasks. As shown in **Table1** and **Table2**, we evaluate CORRECT across **5 models and 9 datasets** areas, and observe consistent and often substantial improvements.(**18.3 → 39.1 on algorithm-generated trajectories,  +46.9 on Math500**). These results indicate that the learned schemata capture broadly useful failure patterns rather than being tied to a specific synthetic setting.
>
> To better address these concerns, we conducted additional experiments in more diverse settings. First, we evaluated CORRECT under a error-category detection dataset **MAST datasets**(see appendix A.13). Even in this coarser setting, CORRECT continues to improve over the baseline, from **11.9 to 15.8 on AG2+GSM8K**. Second, we evaluated CORRECT on previously unseen agentic benchmarks with more heterogeneous environments and task structures. On **AgentErrorBench**, CORRECT improves performance from **36 to 46 on GAIA, from 22 to 35 on ALFWorld, and from 16 to 24 on WebShop**.
>
> We believe these results provide additional evidence that CORRECT is **not limited to one narrow family of synthetic or highly repetitive failures, and that the retrieved schemata can still provide useful guidance in more diverse domains and task formulations.**
>
> **w3/Q1: Broader significance of localized error schemata as reusable memory for continual agent improvement**
>
> We thank the reviewer for this insightful suggestion. We agree that connecting diagnostic credit assignment (error localization) to policy improvement via reusable memory is an exciting direction, and we believe **CORRECT provides a natural foundation for this**.
>
> First, this perspective further highlights a key advantage of CORRECT: it is **training-free and produces explicit, reusable error schemata that can be directly stored, retrieved, and reused across episodes**. Unlike parameter updates, these schemata form a lightweight and interpretable episodic memory, which can be incrementally expanded and refined without retraining.
>
> Second, we would like to clarify that our current evaluation already covers a broad range of settings. In the main paper, we evaluate CORRECT across 5 models and 9 datasets. As shown in **w1/w2**, CORRECT consistently improves performance (**e.g. 18.3 → 39.1 on WHO&WHEN**), suggesting that the learned schemata capture reusable failure patterns across tasks.
>
> Third, to further demonstrate broader applicability, we add new experiments on AgentErrorBench. As shown in **Table 8** and **w1/w2**, CORRECT achieves consistent improvements (e.g., **22 → 35 on ALFWorld**), indicating that the **same schemata-based mechanism generalizes across diverse agentic settings**.
>
> Finally, we agree that bridging error schemata to closed-loop policy improvement (e.g., using schemata as memory to guide future planning or adaptation) is an important future direction. We will incorporate a **dedicated discussion in the final version to outline how CORRECT can be extended toward self-improving agents via episodic memory mechanisms.**
>
> **Q2: Robustness of structural similarity and error localization in long-horizon, noisy MAS trajectories**
>
> Thank you for this important question. **Long-horizon settings are, in fact, one of the main motivations for CORRECT.** In our measured WHO&WHEN Hand-Crafted subset, the trajectories are already very long and noisy: as shown **in Figure 7, about 17.2% of trajectories exceed the largest model context window**, which highlights exactly the challenge the reviewer raises. Despite this, CORRECT still achieves strong improvements on the hardest long-horizon benchmark in Table 1, also as shown in **w1/w2**
>
> More broadly, our method is designed specifically to avoid relying on raw long-context matching. Rather than directly feeding entire long trajectories into the detector, CORRECT retrieves condensed error schemata, which are much more compact and robust to noisy context. Appendix A.10  further shows that schemata retrieval **remains stable under aggressive truncation**, indicating that long trajectories do not pose a practical limitation for the retrieval component itself.
>
> Finally, our cache-size ablation further supports this point. As shown in **Figure 5a**, even with **only 10% of the schemata repository, CORRECT already reaches 69.1% on MMLU-Pro and 38.4% on HotpotQA**, and performance plateaus after only tens to a few hundred schemata. This suggests that, even in long-horizon settings, the recurring structural patterns are concentrated enough that a relatively small schemata pool is sufficient to capture most of them.

---

> > ### Author Rebuttal · Reviewer_KPnZ · 2026-04-04
> >
> > The rebuttal has effectively addressed several of my previous concerns, particularly regarding the potential extension of CORRECT to other agent-related domains. In light of these clarifications, I am glad to raise my rating to 5.

---

> > > ### Author Response · Authors · 2026-04-04
> > >
> > > We sincerely thank the reviewer for recognizing the significance of decisive error recognition in MAS, the key insight that failure patterns exhibit reusable structural recurrence, and the strength of our schema-based design, benchmark construction, and comprehensive empirical analysis. We are also grateful for the insightful and constructive comments, which have helped us further improve the clarity and quality of the manuscript. We will incorporate these suggestions in the final version. Thank you again for your thoughtful review and support. Have a nice day!

---

### Official Review · Reviewer_Prcu · 2026-03-13

**Soundness:** 2
**Presentation:** 3
**Significance:** 2
**Originality:** 2
**Overall Recommendation:** 4
**Confidence:** 3

**Summary:**

The paper proposes CORRECT, a training-free framework for identifying decisive errors in multi-agent system (MAS) execution trajectories. Instead of analyzing full historical logs, the method distills past failures into compact error schemata that capture common patterns and retrieves relevant schemata to guide LLM-based error localization at inference time. The approach enables knowledge transfer across trajectories and avoids expensive fine-tuning. The work also introduces CORRECT-Error, a benchmark containing over 2,000 annotated MAS trajectories generated via an error-injection pipeline and validated by human annotators. Experiments across multiple models and tasks show that CORRECT significantly improves step-level error localization accuracy while introducing minimal runtime overhead.

**Compliance With Llm Reviewing Policy:**

Affirmed.

**Key Questions For Authors:**

Q1. The method relies on LLM-based schema extraction and embedding retrieval. Could the authors clarify the computational overhead of these steps, particularly the additional latency and cost introduced during deployment?

Q2. Since the error schemata are distilled from specific datasets (e.g., WHO&WHEN and CORRECT-Error), could the authors clarify how well the method generalizes to more diverse multi-agent tasks or environments beyond these benchmarks?

A clearer analysis of runtime overhead and a more detailed discussion of cross-domain applicability would strengthen the technical soundness of the submission.

**Limitations:**

yes

**Strengths And Weaknesses:**

**Strengths**

S1. The paper clearly formalizes decisive error recognition in multi-agent trajectories as identifying the earliest step whose correction would change a failure into success. This formulation provides a clear objective for debugging MAS systems and enables precise evaluation of error localization.

S2. The paper introduces a new benchmark, CORRECT-ERROR, which contains more than 2,000 annotated trajectories spanning multiple domains. The dataset is evaluated by human labelers and shows strong inter-annotator agreement on most samples, making it an important benchmark for MAS error recognition tasks.

S3. The work includes thorough ablation studies and detailed training and implementation descriptions (e.g., hyperparameters, prompt templates, and schema extraction procedures), which improves the technical soundness and reproducibility of the work.

**Weaknesses**

W1. A large portion of the benchmark trajectories are synthetically generated. While the authors perform human validation, additional evaluation on real-world multi-agent system trajectories would further strengthen the practicality of the approach.

W2. The failure modes when schema retrieval does not return relevant schemas are not thoroughly analyzed. A deeper discussion of such cases would help understand the robustness limitations of the method.

W3. The cost of schema construction and retrieval during deployment is not fully analyzed. In particular, the additional latency and operational overhead of schema extraction, caching, and matching may affect practicality in real-world systems.

---

> ### Author Rebuttal · Authors · 2026-03-31
>
> **We sincerely appreciate your valuable feedbacks.**
>
> **w1 / q2: Generalization to real-world MAS settings and cross-domain applicability beyond synthetic benchmarks**
>
> We really appreciate this insightful suggestion. In the paper, we have shown that CORRECT achieves competitive results on the human-annotated real dataset of Who and when. Due to the lack of a high-quality human-annotated MAS dataset with step labels, we further evaluated our method on the **MAST dataset**(more details in Appendix A.13), which includes real-world multi-agent system trajectories with human-labeled error categories. On real-world MAST data, CORRECT improves performance from **15.7 → 16.6 (ProgramDev+ChatDev) and 11.9 → 15.8 (Ag2+Gsm8k)**, showing consistent gains in human-labeled MAS settings.
>
> We further evaluate CORRECT on **AgentErrorBench**(more details in Appendix A.13), which contains heterogeneous agentic environments spanning embodied navigation, web interaction, and long-horizon planning. This benchmark differs substantially from CORRECT-Error in both task structure and failure modes. Using GPT-5-nano and tolerant accuracy within ±1 step, CORRECT improves from **36% to 46%** on GAIA, from **22% to 35%** on ALFWorld, and from **16% to 24%** on WebShop. These results suggest that the distilled schemata are not narrowly tied to one synthetic benchmark, but transfer to more diverse tasks, environments, and supervision regimes.
>
> Finally, we also observe **strong cross-model and cross-domain transferability** within our main experiments. As shown in Figure 4a, schemata distilled from one dataset (human-crafted trajectories) generalize effectively to a different distribution (algorithm-generated trajectories) across multiple models (GPT-5-nano, Gemini-2.5-Flash, and Qwen-7B), with **consistent improvements (e.g., 31.8 → 36.5 on Gemini-2.5-Flash)**.
>
> **Together, these results suggest that CORRECT does not rely on dataset-specific patterns, but instead captures reusable structural error patterns that generalize across datasets, models, and environments.**
>
> **w2: Failure modes under missing or irrelevant schema retrieval**
>
> Thank you for this valuable feedback. We want to remark that our model is **robust to retrieval noises where irrelevant schemata are fetched.**
>
> We performed a **hard-negative experiment by injecting random, irrelevant error schemata into the retrieval pool** while keeping the total number of retrieved schemata fixed at 5 to evaluate CORRECT’s robustness to retrieval noise. This simulates cases where top-k retrieval includes partially mismatched or misleading schemata. Even with multiple random schemata injected, CORRECT continues to outperform the baseline on both model families. For example, with 5 random schemata, CORRECT still achieves **10.7 vs. 8.6 on GPT-5 and 6.9 vs. 3.5 on Qwen-7B**. With 1 random schema, performance remains strong at **13.8 on GPT-5 and 10.7 on Qwen-7B**, compared to full CORRECT scores of **17.2 and 12.3**, respectively. While accuracy decreases as noise increases, the method remains consistently above baseline, demonstrating strong robustness to retrieval noise and hard negatives. More details are provided in Appendix A.14
>
> Beyond this worst-case setting, we also analyze coverage via cache size. In Figure 5a, with only **10%** of the schemata repository, CORRECT already reaches 69.1% Acc@1 on MMLU-Pro and 38.4% Acc@1 on HotpotQA, and performance plateaus beyond 50% of the repository (i.e., only tens to a few hundred schemata) . This suggests that, in practice, **a relatively small cache already covers most recurring error patterns, making complete retrieval failure unlikely**.
>
> Overall, these results suggest that **when retrieval fails, CORRECT degrades gracefully toward a weak-schemata or near-zero-shot regime, rather than catastrophically failing, and remains robust both to noisy retrieval and limited schemata coverage.**
>
> **w3 /q1: Computational overhead and deployment cost of schema extraction and retrieval**
>
> Thank you for the insightful question. We want to remark that CORRECT is a **lightweight framework** that introduces **negligible overhead**. During online inference, CORRECT only retrieves the most relevant schemata from the cache and appends them to the prompt, which adds less than **5%** overhead to both latency and API cost. Concretely, the API cost increases only from **$0.86 to $0.89**, and the inference time increases only from **44s** to **47s**.
>
> Moreover, schemata construction is performed offline and amortized across many requests, rather than being paid repeatedly at serving time. As a result, the practical deployment overhead is dominated by the lightweight online retrieval step above. **These results suggest that CORRECT is practical for real-world systems, as it achieves substantial accuracy gains while introducing only minimal operational cost.**

---

> > ### Author Rebuttal · Reviewer_Prcu · 2026-04-03
> >
> > Thanks for your response.
> >
> > **Regarding the failure modes**
> >
> > I appreciate the hard-negative experiment and agree that it provides useful evidence of robustness. However, it does not fully address my original concern regarding the system’s behavior when retrieval fails or returns incorrect schemata. Could you clarify more explicitly what happens in such scenarios? For example, when the retrieved schemata are irrelevant, partially mismatched, or misleading, how does the downstream reasoning process change? A more concrete analysis of these cases would help delineate the robustness boundaries of the method.
> >
> > **Regarding the deployment cost**
> >
> > Let me restate my question more precisely.
> >
> > In Section 5.2, the cache-size analysis is presented as a percentage of the full schema library. My first concern is the concrete scale in practice: how large is the schema library, and what do these percentages correspond to in absolute terms?
> >
> > In addition, since retrieval is a central component of the framework, I am particularly interested in the latency and cost of the retrieval step itself, specifically the semantic similarity search described in Section 3.2.
> >
> > A clearer quantification of these aspects would allow readers to better assess the practical overhead and real-world applicability of the proposed approach.

---

> > > ### Author Response · Authors · 2026-04-04
> > >
> > > Thank you very much for the clarification and the constructive feedback!
> > >
> > > CORRECT is explicitly designed to be **robust to imperfect or even misleading retrieval**, and we provide both **empirical and design-level evidence**.
> > >
> > > First, CORRECT retrieves **top-k schemata** rather than a single match, which **reduces over-reliance on any individual schema**. More importantly, our hard-negative experiment shows that even when injecting random irrelevant schemata, performance remains above the no-schema baseline (e.g., GPT-5 improves from 8.6 → 15.5), indicating that **misleading schemata do not dominate the final decision**.
> > >
> > > Second, the results in **Tables 5–6** further suggest that performance is **not sensitive to precise matching**, but instead depends on **overall cache quality**. For example, varying the refinement threshold improves performance from 80.4 → 81.3, and adjusting similarity thresholds yields gains from 76.5 → 78.9, even though these changes **do not modify the retrieval** mechanism itself. This indicates that CORRECT does not rely on exact matches, but instead **benefits from a diverse and well-structured schema pool.**
> > >
> > > Finally, CORRECT includes a schema expansion mechanism: when no sufficiently similar schema exists, a new schema is added to the cache. Combined with distillation, this ensures that **coverage and quality improve over time**, making complete retrieval failure increasingly unlikely in practice.
> > >
> > > Regarding scale and retrieval cost, the schema cache is small in practice. For example, on HotpotQA, the full cache contains **572 schemata (10% ≈ 57)**, and for MMLU, we use CORRECT-Error with **2264 schemata (10% ≈ 226)**. Retrieval is highly efficient: the average retrieval latency is around **35ms hosted with Faiss-GPU [1] version**. We remark that the overhead can be further optimized with complementary optimizations. For instance, FusionANNS[2] achieves retrieval with **~2–4 ms latency at 9K–21K QPS on a V100 GPU at billion-scale datasets.**
> > >
> > > As a result, the end-to-end overhead is minimal. Compared to a 44s baseline, CORRECT takes 47s total, where retrieval contributes **~35 ms** and generation overhead ~3s. This confirms that CORRECT **remains lightweight in practice, with overhead dominated by the base LLM call**.
> > >
> > > Finally, we emphasize reproducibility: *We have already included the hyperparameters, prompts, and detailed pseudo-codes, see Algorithm1*, in the paper. We will release *CORRECT-Error, schema construction pipelines, retrieval code, and all prompts and hyperparameters to facilitate adoption*.
> > >
> > > [1] ```https://github.com/facebookresearch/faiss/wiki/Faiss-on-the-GPU```
> > >
> > > [2] Tian, Bing, et al. "Towards High-throughput and Low-latency Billion-scale Vector Search via {CPU/GPU} Collaborative Filtering and Re-ranking." 23rd USENIX Conference on File and Storage Technologies (FAST 25). 2025.

---

### Official Review · Reviewer_AfJK · 2026-03-13

**Soundness:** 3
**Presentation:** 3
**Significance:** 3
**Originality:** 3
**Overall Recommendation:** 4
**Confidence:** 4

**Summary:**

The authors present CORRECT, a training-free framework for localizing decisive errors in Multi-Agent Systems (MAS). Finding the exact step where an error originates in MAS is notoriously difficult because failures cascade across multiple agents and the execution logs can easily exceed context windows. Instead of relying on fine-tuning or naive in-context learning, the proposed method distills past failure logs into compact "error schemata" (which capture signatures, contexts, and heuristics). These schemata are stored in an online cache and retrieved at inference time based on semantic similarity to guide an LLM judge. To evaluate their approach, the authors also introduce CORRECT-Error, a dataset of over 2,000 annotated failure trajectories created through a synthetic error-injection pipeline. They report localization accuracy improvements of up to 19.8% over existing methods.

**Compliance With Llm Reviewing Policy:**

Affirmed.

**Key Questions For Authors:**

1. How does the system perform when the retrieval mechanism encounters structurally similar errors that have vastly different semantic vocabularies?

2. Did you evaluate the frequency of hallucinated or incorrect detection heuristics generated during the offline schema extraction phase?

3. What is the actual absolute accuracy of CORRECT on the hardest datasets, and do you beleive LLM-as-a-judge can ever reach production level reliability for this task?

**Limitations:**

The authors discuss the challenges of MAS debugging, but they need to explicitly address the vulnerability of their cache to poisoned or hallucinated schemata generated during the offline phase. If an LLM writes a bad heuristic, the system has no built-in self correction mechanism to prune it out of the cache.

**Strengths And Weaknesses:**

Strengths:

- Bypassing the massive context limits of MAS execution logs by distilling them into reusable schemata is a highly practical systems engineering choice. It avoids the computational overhead of fine-tuning.

- The introduction of the CORRECT-Error dataset is a good addition. The community needs larger scale, step-level annotated benchmarks for multi-agent debugging, and the synthetic error-injection pipeline is a good way to scale data generation.

- The framework's near-zero runtime overhead makes it a very appealing plug-and-play solution for production deployments.

Weaknesses:

- The reliance on BERT-based embeddings for both offline trajectory clustering and online schema retrieval is a notable bottleneck. Execution logs are noisy, and semantic similarity might completely miss structural error equivalences if the domain vocabulary differs. A purely semantic retriever will likely struggle with out-of-distribution applications.

---

> ### Author Rebuttal · Authors · 2026-03-31
>
> **We sincerely appreciate your valuable feedbacks.**
>
> **w1 / Q1: Robustness of schema retrieval under OOD and noisy retrieval**
>
> We note that our method can **generalize robustly across Datasets and Models**. Figure 4a shows that schematas distilled from human-crafted trajectories transfer effectively to algorithm-generated data. Across GPT-5-nano, Gemini-2.5-Flash, and Qwen-7B, CORRECT with transferred schematas consistently outperforms baselines, with Gemini-2.5-Flash improving **31.8–36.5%**. This cross-domain transferability indicates that distilled schematas capture fundamental error patterns rather than context specific patterns.
>
> We further evaluate **worst-case retrieval mismatch via hard-negative experiments** (Appendix A.14), where random irrelevant schemata are injected. Even with 5 random schemata, CORRECT remains above baseline (e.g., GPT-5: **10.7** vs. **8.6**; Qwen-7B: **6.9** vs. **3.5**), showing that **mismatched schemata do not actively hurt and can be filtered by the model**.
>
> **Q2 Robustness to hallucinated / incorrect schemata**
>
> First, we emphasize that our evaluation **directly uses the automatically generated schemata without manual filtering, including potentially imperfect or hallucinated detection heuristics**. Despite this, CORRECT still achieves substantial gains over baseline. For example, on the WHO&WHEN benchmark, GPT-5 improves from **8.6** to **17.2** on the human-crafted subset and from **18.3** to **39.1** on the algorithm-generated subset, showing that raw generated schemata are already effective in practice.
>
> Second, our experiments provide quantitative evidence that the automatically generated schemata are robust and broadly useful in practice. As discussed in Q1 and shown in Figure4a, our method  **generalize robustly across Datasets and Models**.
>
> Relatedly, Figure 4b shows that improving the **offline schemata generator can further raise downstream detection accuracy**: for example, using GPT-5 instead of Qwen-72B for schemata generation increases accuracy from **8.6 to 12.2** for GPT-5-nano and from **10.3 to 12.2** for Qwen-2.5-7B . We view this as complementary evidence: the base schemata are already robust enough to transfer across settings, and higher-quality generators can further refine them.
>
> Third, CORRECT includes an **explicit refinement mechanism for imperfect schemata**. The paper states in Algorithm 1 that frequently accessed schemata are re-generated and replay-evaluated, and the best candidate replaces the old one (Lines 21–25) . Consistent with this design, the paper reports in Table 6 that increasing the hotness threshold refinement yields small but consistent gains, e.g. **ARC improves from 80.4 to 81.3 as θ_hot increases from 0 to 0.3**.
>
> Fourth, CORRECT is **robust even when retrieval includes clearly bad schemata**. As shown in w1/Q1, CORRECT remains robust even under severe retrieval noise: with 5 random schemata, performance stays above baseline (GPT-5: 10.7 vs. 8.6; Qwen-7B: 6.9 vs. 3.5), indicating that noisy or mismatched schemata do not substantially harm inference.
>
> **Overall, Figure 4a, Figure 4b, Algorithm 1, Table 6, and Tables 9–10 together provide concrete evidence that schemata quality matters, that imperfect schemata can be refined automatically, and that the system remains robust even under substantial schemata noise**.
>
> **Q3 + Limitation: Absolute performance, scalability, and cache robustness**
>
> We appreciate this concern about the poisoned or hallucinated cache. **CORRECT does not treat the offline cache as static after schemata generation**. It has two built-in safeguards: **selective expansion and schemata distillation**. First, new schemata are added only when an incoming failed trajectory is sufficiently dissimilar to the existing cache, controlled by the similarity threshold δ. This prevents redundant or low-value schemata from accumulating blindly. Second, frequently accessed schemata are revisited through distillation: schemata whose access frequency exceeds the hotness threshold θ_hot are re-generated and replayed on prior trajectories, and the most discriminative schemata is retained. In other words, the cache is designed to be updated and compacted over time rather than permanently preserving every initial heuristic.
>
> We also provide empirical evidence that this mechanism is stable. In the revised appendix, varying θ_hot produces small but consistent gains on ARC **(80.4 → 80.6 → 81.2 → 81.3 for θ_hot = 0.0, 0.1, 0.2, 0.3)**, which suggests that replay-based refinement is helpful rather than brittle. Similarly, the similarity threshold δ mainly trades off cache compactness against accuracy **(76.5, 78.1, 78.9 for δ = 0.6, 0.7, 0.8)**, indicating that CORRECT remains stable across reasonable cache-management settings.

---

> > ### Author Rebuttal · Reviewer_AfJK · 2026-04-06
> >
> > Thank you for your rebuttal, I increase my score to 5.

---

> > > ### Author Response · Authors · 2026-04-06
> > >
> > > We sincerely thank you for recognizing the practical value of CORRECT and for your insightful feedback. We will definitely incorporate your suggestions into the final version.
> > >
> > > We also greatly appreciate your note about increasing the score to 5. At the moment, the updated score is not yet visible on our side, so we would be grateful if you could kindly confirm that the change has been saved in the review system when convenient.
> > >
> > > Thank you again, and have a nice day!

---

### Decision · Program_Chairs · 2026-04-30

**Decision:**

Accept (regular)

**Comment:**

This paper introduces a training-free framework for decisive error recognition in multi-agent systems, distilling past failure trajectories into compact, reusable error schemata that are retrieved at inference time to guide an LLM-as-judge. The central insight that failures in MAS exhibit strong structural recurrence, is well-motivated and empirically supported, leading to a clean and coherent design. The paper’s contributions include the CORRECT framework, the CORRECT-Error benchmark, and strong empirical performance gains.

The rebuttal was highly effective. Three of the four reviewers raised their scores to 5 (one only reflected in comment), and the authors provided additional experimental results that materially strengthened the submission. For the final version, the authors need to make sure to include the naive-ICL vs. CORRECT ablation and the decisive-intervention results in the main text, as well as a discussion of limitations.

Overall, I recommend acceptance.